# *In silico* identification of natural product inhibitors against Octamer-binding transcription factor 4 (Oct4) to impede the mechanism of glioma stem cells

Chirasmita Nayak, Sanjeev Kumar Singh [ID]*

Computer-Aided Drug Design and Molecular Modeling Lab, Department of Bioinformatics, Alagappa University, Karaikudi Tamil Nadu, India

* skysanjeev@gmail.com

**Data Availability Statement:** All data are fully available in the MS.

**Funding:** The author(s) received no specific funding for this work.

## Abstract

Octamer-binding transcription factor 4 (Oct4) is a core regulator in the retention of stemness, invasive, and self-renewal properties in glioma initiating cells (GSCs) and its overexpression inhibits the differentiation of glioma cells promoting tumor cell proliferation. The Pit-Oct-Unc (POU) domain comprising POU-specific domain (POU$_S$) and POU-type homeodomain (POU$_{HD}$) subdomains is the most critical part of the Oct4 for the generation of induced pluripotent stem cells from somatic cells that lead to tumor initiation, invasion, posttreatment relapse, and therapeutic resistance. Therefore, the present investigation hunts for natural product inhibitors (NPIs) against the POU$_{HD}$ domain of Oct4 by employing receptor-based virtual screening (RBVS) followed by binding free energy calculation and molecular dynamics simulation (MDS). RBVS provided 13 compounds with acceptable ranges of pharmacokinetic properties and good docking scores having key interactions with the POU$_{HD}$ domain. More Specifically, conformational and interaction stability analysis of 13 compounds through MDS unveiled two compounds ZINC02145000 and ZINC32124203 which stabilized the backbone of protein even in the presence of linker and POU$_S$ domain. Additionally, ZINC02145000 and ZINC32124203 exhibited stable and strong interactions with key residues W277, R242, and R234 of the POU$_{HD}$ domain even in dynamic conditions. Interestingly, ZINC02145000 and ZINC32124203 established communication not only with the POU$_{HD}$ domain but also with the POU$_S$ domain indicating their incredible potency toward thwarting the function of Oct4. ZINC02145000 and ZINC32124203 also reduced the flexibility and escalated the correlations between the amino acid residues of Oct4 evidenced by PCA and DCCM analysis. Finally, our examination proposed two NPIs that can impede the Oct4 function and may help to improve overall survival, diminish tumor relapse, and achieve a cure not only in deadly disease GBM but also in other cancers with minimal side effects.

**Competing interests:** The authors have declared that no competing interests exist.

**Abbreviations:** GBM, Glioblastoma; GSCs, glial stem cells; IFD, Induced fit docking; iPSC, induced pluripotent stem cell; MDS, Molecular Dynamics Simulation; Oct4, Octamer-binding transcription factor 4; POU, Pit-Oct-Unc; RBVS, Receptor based virtual screening.

## Introduction

Glioblastoma Multiforme (GBM) is a common and devastating malignant form of high-grade glioma accounting 70% of central nervous system tumors [1, 2]. Despite surgical resection in combination with radiation therapy and adjunct temozolomide (TMZ) chemotherapy, the clinical prognosis and median overall survival of GBM patients remain grim with 12–14 months and 5-year survival with3-5% after initial diagnosis [3]. Despite its relatively low frequency, GBM is responsible for 4% of all deaths caused by cancer [4]. Recurrence of malignant tumors and therapeutic resistance within months following adjuvant therapy are the key contributors to GBM patient's death and the major challenges in treatment [5]. Evidence signposts from some studies exposed that a small group of cells called "glioma stem-like cells" (GSCs) underlie tumor propagation, drug resistance, and relapse after conventional therapy [6]. GSCs exhibit a quiescence and self-renewal capacity which is responsible for tumor heterogeneity, maintenance, and metastasis [7]. Therefore, there is an urgent need for specific targeted therapies aiming at the elimination of GSC which could beat this devastating disease by rendering inordinate significance to the therapeutic advancement of GBM.

Octamer-binding transcription factor 4 (Oct4) is well known as a core GSC regulator establishing a critical role in GSC maintenance including self-renewal, pluripotency, reprogramming, cellular plasticity, and repressing differentiation [8, 9]. Oct4 belongs to POU (Pit-Oct-Unc) transcription factor and comprises three domains namely (1) POU domain for DNA binding (2) an N-terminal transactivation domain and (3) a C-terminal domain which acts as cell type-specific transactivation domain [10]. The POU domain is composed of two subdomains i.e. POU-specific domain (POU$_S$) with four alpha-helices and the POU-type homeodomain (POU$_{HD}$) domain with three alpha-helices tethered by a flexible linker (approximately 17 amino acid residues long) region [11, 12]. The helix of POU$_S$ and the N-terminal part of the POU$_{HD}$ domain can interact with major groove and minor groove of DNA, respectively, thereby triggering downstream gene transcription [13]. Oct4 only the member from the POU family that plays a pivotal role in stem cell renewal and pluripotency; and cannot be replaced by any other POU family to reprogram the induced pluripotent stem cell (iPSC) [14]. Expression of Oct4 is regarded to be restricted to pluripotent stem cells and its expression rapidly down-regulated upon initiation of differentiation [15]. In recent years, a series of studies reported that Oct4 is highly expressed in various benign and malignant tumors, including glioma, lung cancer, bladder cancer, breast cancer, pancreatic cancer, hepatoma, and oral cancer [8, 16]. In primary glioma, Oct4 was highly up-regulated and the expression levels were increased in parallel with pathological grading [8, 17].

In some studies, it has been demonstrated that Oct4 cooperatively activated the SOX2 (sex-determining region Y-box 2) enhancer region by interacting with SOX4 to maintain the stemness properties of GSCs [18]. Further, Oct4 upregulated the phosphorylation of Stat3 to proliferate the tumor cells and increased the expression of Nestin to inhibit the differentiation of glial cells [17]. Together with NANOG, SOX2, and other transcription regulators, Oct4 activated both protein-coding genes and non-coding RNAs necessary for pluripotency which correlated with cell fate determination, proliferation, metastasis, drug resistance and invaded from apoptosis in cancer cells [19, 20]. Oct4 maintained the cancer cell survival partly by inhibiting apoptosis through the Oct4/TCL1/AKT1 pathway where Tcl1 enhances the kinases activity of Akt1 to promote cell proliferation [21]. Most importantly, it has been reported that the overexpression of the only Oct4 found to be a crucial factor for pluripotent of cancerous cells including GBM [18], lung cancer [22], human oral squamous cell carcinoma, bladder cancer, and seminoma cancer [21]. Downregulation of Oct4 resulted in the reduction of glioma colony formation and cell proliferation *in vitro* and *in vivo* which indicates the vital role

of Oct4 in tumorigenic activity in GBM self-renewal, aggressiveness, and pluripotency [17, 18].

Although Oct4 plays a pivotal role in the maintenance of pluripotency, stemness, aggressiveness, no small molecule inhibitors have been discovered yet to impede the function of Oct4. Therefore, we hunt for potent small molecule inhibitors to abolish the function of Oct4 using *in silico* approach which can increase the survival rate not only in GBM patients but also in most cancer patients. In this study, we investigated the disordered regions present in Oct4 using various tools followed by STRING analysis to explore the functional interactive networks with other proteins. Subsequently, receptor-based virtual screening was performed to retrieve potential leads from various natural product databases. Finally, the conformational and interaction stability of the complexes was evaluated through molecular dynamics simulation. From this study, we identified two compounds showing stable interaction with Oct4 even in the presence of the linker region and in dynamics conditions.

## Materials and methods

### Sequence analysis

The reviewed full-length protein sequence (UniProt ID: Q01860) of Oct4 was retrieved from the UniProtKB sequence database [23] representing 360 amino acids (AA). The obtained sequence was searched through DISPRED3 [24], IUPred2A [25], and PONDR [26] to analyze the residue level of disordered propensity. In PONDR, the VL-XT predictor algorithm was used to predict the disorder regions which uses three integrated feedforward neural networks: the VL1 predictor [27], the N-terminus predictor (XN), and the C-terminus predictor (XC) [28]. Subsequently, the updated version of STRING (Search Tool for the Retrieval of Interacting Genes/Proteins) v11.0 [29] was employed to construct the functional interaction associative network and interaction of Oct4 with other proteins. Further, physicochemical properties including molecular weight, isoelectric point, instability index, aliphatic index, and grand average hydropathicity (GRAVY) were computed by the ProtParam too [30] of Expasy Proteomic Server [31].

### Protein structure prediction and preparation

Since the structure of human Oct4 has not yet been solved experimentally, comparative modeling was executed using MODELLERv9.19 [32] to construct 50 3D models. The 3D coordinates were generated based on the crystal structure of Mus Musculus Oct4 (PDB ID: 3L1P) [33] with 42% identity and 87.77% query coverage (For full length). The generated models were ranked according to discrete optimized potential energy (DOPE) and the lowest energy model was selected as the best model. Optimization followed by energy minimization of the best model was performed by Protein preparation wizard until the root mean square deviation (RMSD) of the non-hydrogen atoms touched 0.3Å by applying OPLS-AA force filed [34, 35]. The minimized structure was evaluated using ProSA, and Saves server to check overall potential errors, dihedral angle distribution, and calculate the non-bonded interactions between the atoms respectively [36, 37]. The validated model was superimposed with the template structure to investigate the reliability of the structure.

### Molecular dynamics simulation

MD simulation was carried out to comprehend the stability and dynamic behavior of modeled structure using Desmond with Optimized Potentials for Liquid Simulations (OPLS) 2005 force field [38]. The protein was soaked with TIP3P as solvent inside an orthorhombic box and

neutralized by adding appropriate counter ions and 0.15M of salt concentration [39]. The distance between the protein and box wall was set to 10Å to avoid the steric interaction with its periodic image. Each simulation was subjected to energy minimization applying a hybrid method of steepest descent and the Limited memory Broyden–Fletcher–Goldfarb–Shanno (LBFGS) algorithms with maximum 5000steps until a gradient threshold was touched up to 25 kcal/mol [40]. The systems were relaxed by constant NVT (number of particles, Volume and Temperature) ensemble conditions for 1ns to produce simulation data for post-simulation analysis [41, 42]. The temperature was maintained at 300K for whole simulations using Nose-Hoover thermostats and the Martyna-Tobias-Klein barostat method was used to maintain stable pressure. To examine the equation of motion in dynamics, a multi-time step RESPA integrator algorithm was used [43]. The final equilibrated system was carried to perform a 100ns molecular dynamics simulation and the result was analyzed through the event analysis module of Desmond.

## Binding site prediction and grid generation

The Druggability sites were identified through SiteMap implemented in Schrodinger [44]. SiteMap predicts the binding sites using Goodford's GRID algorithm which locates the energetically favorable sites by using the interaction energies between the protein and grid probes [45]. The druggable sites were predicted by various physical descriptors such as size, the degree of enclosure, the degree of exposure, tightness, hydrophobic, hydrophilic, hydrogen bonding possibilities, and linking site points that promote protein-ligand interaction [46]. The sites were then ranked based on the SiteScore (Eq 1) and Dscore (Eq 2) respectively.

$$SiteScore \; = \; 0.0733n1/2 + 0.6688e \; - \; 0.20p \tag{1}$$

$$Dscore \; = \; 0.094n1/2 + 0.60e \; - \; 0.324p \tag{2}$$

Where n is the number of site points, e is the enclosure score, and p is the hydrophilic score [47]. After identification of ranked potential druggability sites, the sites were validated by superimposing the modeled structure with the crystal structure (PDB ID: 3L1P) and the best site was chosen for further study.

The validated and top-ranked druggability site predicted by SiteMap was prearranged as Glide input files for the generation of the receptor grid [48]. The white sphere of SiteMap was picked to distinguish the position of ligand applying van der Waals radius scaling factor of 1.0 and partial charge cutoff of 0.25 [49].

## Ligand database preparation and ADME screening

Initially, the complete ZincNPD, NCINPD, and NPB databases were prepared using the LigPrep module [50]. The optimized potential for the liquid simulation (OPLS)-2005 force field was employed to preserve the original state and the chirality of ligands [51]. Further, the Absorption Distribution Metabolism and Elimination (ADME) properties were evaluated by employing the QikProp module of Schrodinger suite [52] to assess the druggability and filter the drug-like molecules at an early stage before identifying the new inhibitors. QikProp predicts both physicochemical and pharmacokinetic properties of the compounds [53]. For considering the compound in the present study, the compound must satisfy the following criteria

1. Molecular weight of the molecule (**mol_MW**)    130.0–725.0

2. Predicted central nervous system activity on a –2 (inactive) to +2 (active) scale (**CNS**) –1 to +2

3. Hydrogen bond donor (**donorHB**)    0.0–6.0

4. Hydrogen bond accepter (**accptHB**)    2.0–20.0

5. Predicted octanol/water partition coefficient (**QPlogPo/w**)    −2.0–6.5

6. Predicted aqueous solubility (**QPlogS**)    −6.5–0.5

7. Predicted brain/blood partition coefficient (**QPlogBB**)    −3.0–1.2

8. Predicted human oral absorption on 0 to 100% scale (**PercentHuman-OralAbsorption**) 25% is poor >80% is high

## High throughput virtual screening

To identify novel potent inhibitors that interact with the druggable regions of Oct4, receptor-based virtual screening (RBVS) was performed using the Virtual Screening Workflow (VSW) available in Glide, Schrodinger [54]. Glide utilizes Systematic and Simulation methods to discern the conformations and flexibility of ligands [54]. The processed compounds were elapsed through Glide based three stages of docking protocol such as High Throughput Virtual Screening (HTVS), Standard Precision (SP), and Extra precision (XP) [46]. In the first phase, HTVS docking retrieved 10% of the top compounds based on their scoring values where fewer scoring compounds were eliminated. The HTVS retrieved compounds were subjected to SP docking as input files and the top 10% screened compounds were passed through XP docking. Glide XP mode determines all reasonable conformations for each low energy conformer in the designated binding site [55]. The Glide scoring function (G-score) was applied to evaluate the final energy and to pick the best conformation for each ligand during the docking procedure. GScore is a modified and extended version of the empirical scoring function that combines various parameters [56, 57]. The GScore is calculated in Kcal/mol as

$$GScore = a*vdW + b*Coul + Lipo + Hbond + Metal + Site + BuryP + RotB(3)$$

Where vdW = Vander-Waals forces, Coul = columbic forces, Lipo = hydrophobic interactions, Hbond = Hydrogen bonds, Metal = metal-binding term, Site = polar interactions in the binding site, Bury P = penalty for the buried polar group, RotB = freezing rotatable bonds and the coefficients of vdW and Coul are a = 0.065 and b = 0.130.

## Induced fit docking

Induced fit docking (IFD) was performed to understand the specific interaction between the modeled structure of Oct4 and screened compounds using IFD application in Schrodinger [58]. IFD uses both rigid receptor docking (using Glide) and protein structure prediction and refinement (using Prime) code to provide the best conformation. Initially, Glide [59] generates possible conformations of the ligand and the top 20 poses of each ligand were refined by Prime refinement module [60] via rotamer-based library optimizations of the protein side-chain conformations [61]. Finally, the putative docked complexes and poses were ranked using the Glide score function and prime where the best complexes were chosen based on Glide docking Score, energy, and visual inspection [62, 63].

## Binding free energy calculation

The binding free energies of the putative complexes were calculated through prime-MM/GBSA (Molecular Mechanics/Generalized Born Surface Area) [60] to cross verify the docking

results of Glide XP and IFD. MM/GBSA is an elision of a procedure that integrates OPLS molecular mechanics energies ($E$MM), an SGB solvation model for polar solvation ($G$SGB), and a nonpolar solvation term ($G$NP) comprised of the nonpolar solvent accessible surface area and van der Waals interactions [64]. The binding free energy calculation is expressed as

$$\Delta G\text{bind} \; = \; G\text{complex} \; - \; (G\text{protein} \; + \; G\text{ligand}) \tag{4}$$

Where

$$G = E\text{MM} + G\text{SGB} + G\text{NP} \tag{5}$$

The Gaussian surface area model instead of vdW was engaged by Prime for representing the solvent-accessible surface area [65].

## Molecular dynamics simulation

To investigate different conformation and interaction of the compounds within the binding pocket of Oct4, molecular dynamics simulation was carried out using Desmond [38]. The docking complexes were used as initial structures for computing 50ns MD simulation. The structures were imported in the set-up wizard of Desmond and soaked with TIP3P inside an orthorhombic box. All other steps were followed as mentioned in the free form of Oct4 protein simulation. ProDyv1.10.10 was used to perform Principal component analysis (PCA) followed by dynamic cross-correlation matrix (DCCM) and prody interface of the VMD-integrated Normal Mode Wizard(NMW) was employed to construct porcupine plots [66].

# Results and discussion

## Identification of disordered region and *in silico* characterization

To analyze the propensity for the intrinsically disordered region of Oct4, the obtained precise sequence (UniProt ID: Q01860) was subjected to PONDR VL-XT, DISOPRE3, and IUPred2A. Disordered regions (DRs) were defined as the protein or some part of proteins that lack fixed or ordered three-dimensional structures that can facilitate their interactions with other proteins and allow more structural changes in the modeled structure [67, 68]. Fig 1 illustrated the overall disorder profile of the Oct4 protein. According to PONDR VL-XT, Oct4 contains 163 (from 360) disordered residues that are characterized as 45.28% of overall disorder probability. Further, PONDR VL-XT predicted that there are six DRs consist of at least 15 residues (residues 1–26, 42–59, 81–126, 218–243, 275–303 and 340–360). The dark black box shown in Fig 1a represented the disorder binding region which can get stable order structure during protein-protein interactions [69, 70]. DISOPRED 3 predicted the high confidence disorder region at the C-terminal of Oct4 comprising approximately 87 residues long (residues 275–360). DISOPRED 3 also determined three disorder-to-order transition regions (residues 1–21 of N-terminal, 210–230 and 280–360 of C-terminal) which can be stable structure upon binding with other proteins (Fig 1b). IUPred2A illustrated that the N-terminal and C-terminal are disorder regions and Oct4 contains a sensitive disorder region (residues 59–79) that can undergo disorder-to-order transition to form a stable conformation (Fig 1c) [25]. PONDR VL-XT and DISOPRED 3 showed a similar pattern of disorder region prediction except at the C-terminal whereas IUPred prediction varies from N-terminal to C-terminals. Despite their different algorithm to predict the disorder regions, all servers categorized the approximately first 25 amino acid residues of the N-terminal, 77–125 amino acid residues and 286–300 residues of the C-terminal region of Oct4 as intrinsically disordered region (Fig 1d).

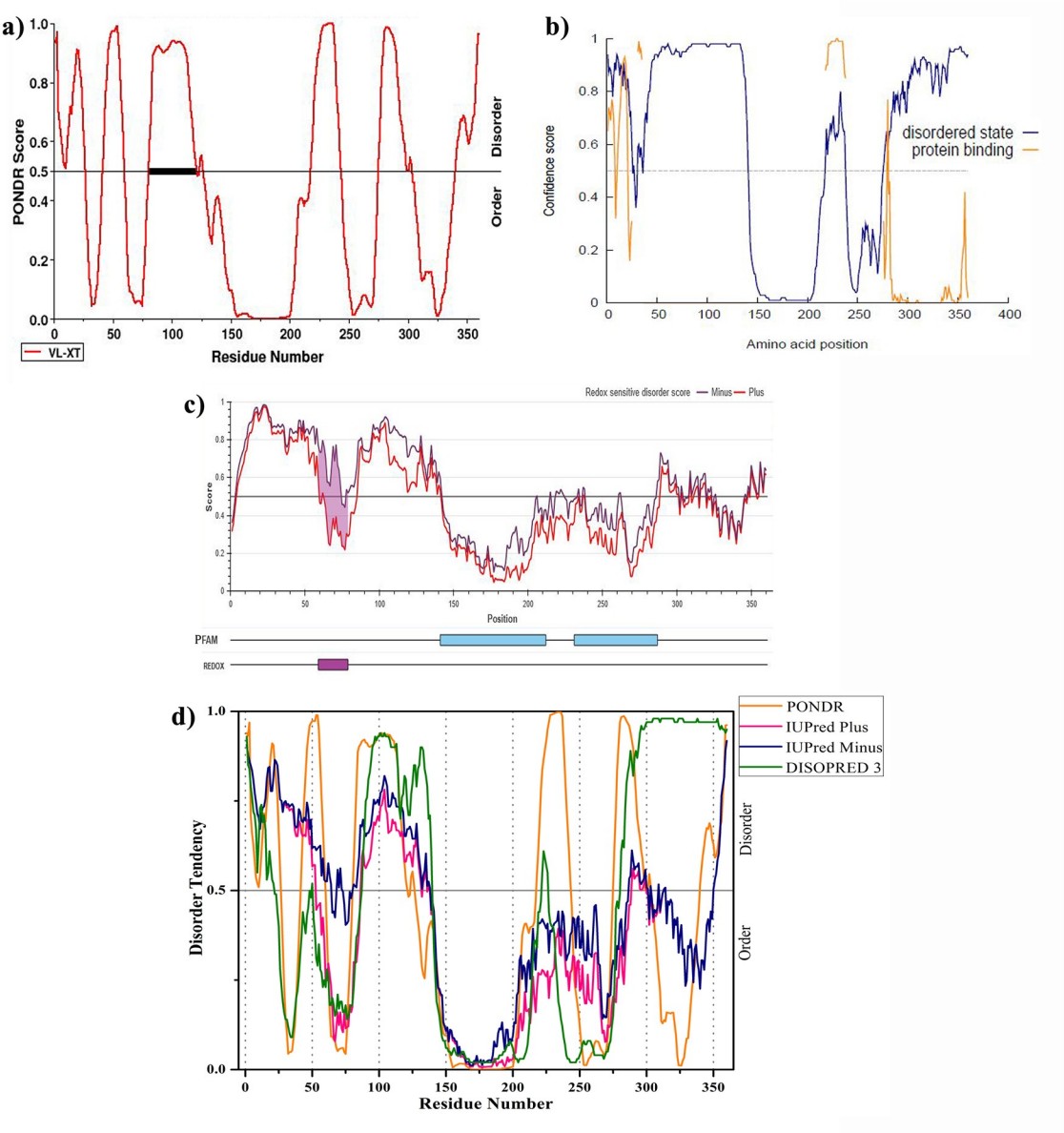

**Fig 1. Intrinsically disordered predisposition of Oct4 protein.** a) PONDR VL-XT b) DISOPRE3 c) IUPred2A d) comparison of three predicted results.

Protein interaction patterns of Oct4 were investigated through the STRING database using the sequence search tool and the interaction map illustrated in Fig 2. The generated interaction diagram from STRING disclosed that Oct4 interacts with SOX2, Nanog, KLF4, ZSCAN10, CDX2, EPAS1, FOXD3, and so on (Table 1). Oct4 heterodimerizes with SOX2 and generates induced pluripotent stem cells from differentiated cells [71, 72]. Oct4 together with other transcription factors viz. KLF4, ZSCAN10 possesses a magic power to reprogram the pluripotency in differentiated cells; to maintain embryonic stem cells, proliferation, self-renewal features and prevent cell differentiation [73, 74]. Oct4 cooperating with SOX2 and NANOG regulates several gene expression where these genes play a key role in signaling pathways that attribute

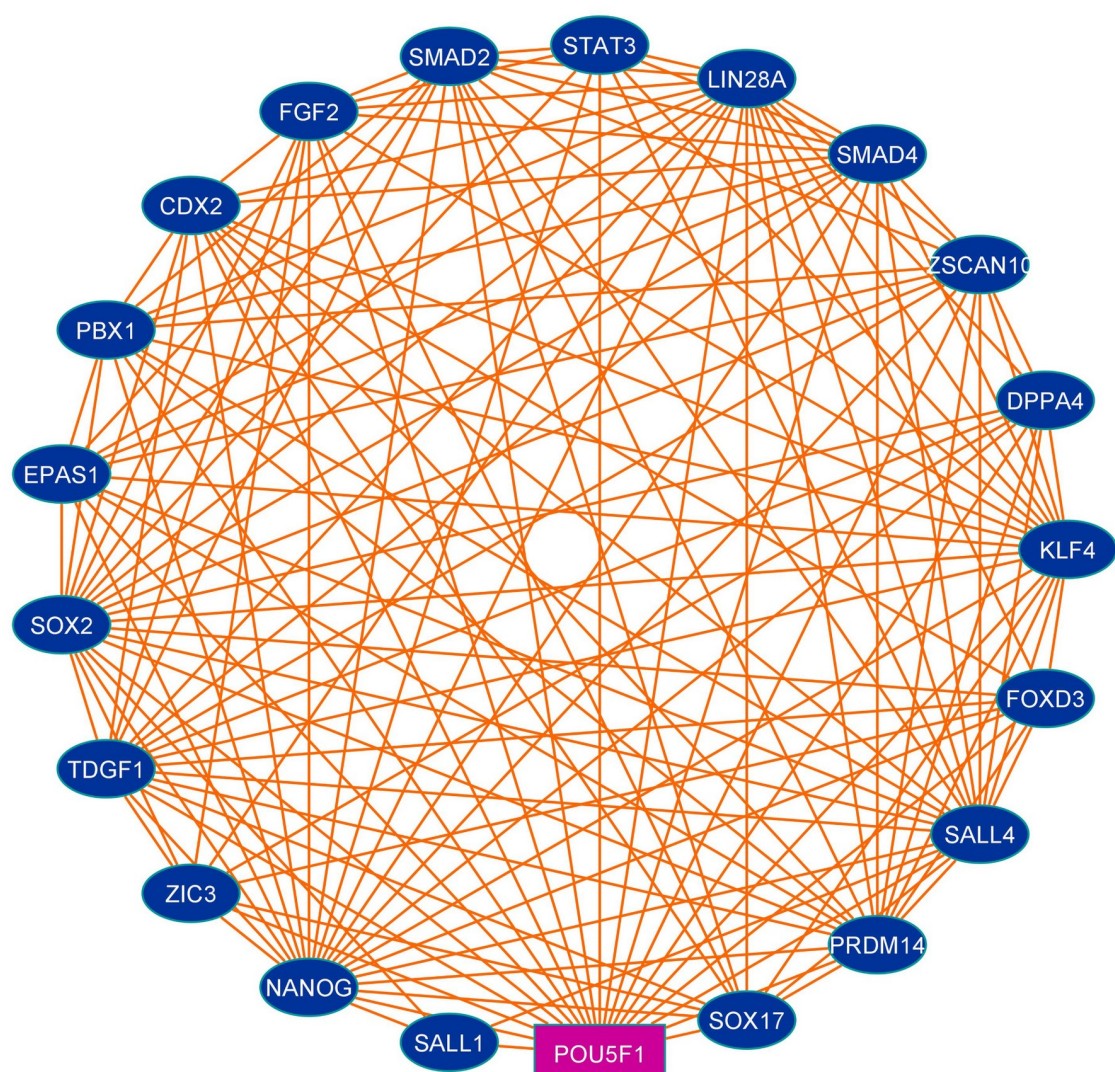

**Fig 2. Protein-protein interaction map of Oct4 generated using online tool STRING.** The pink colored rectangular node represents the query protein (Oct4).

pluripotency, reoccurrence, and self-renewal of tumor cells [19, 72]. FOXD3 suppresses the interneuron differentiation by restricting the neural progenitor cells to the lineage neural crest and maintain the pluripotency [75]. FOXD3 controls the expression of Oct4, NANOG, and SOX2 and participates in the maintenance of stemness, pluripotency, and survival of precursor cells [76]. EPAS1 promotes the Oct4 expression which regulates the maintenance, differentiation, and survival of stem cells thereby contributing to tumor-promoting activity [77]. Moreover, the interaction network result proposed that Oct4 communicated with mostly transcription factors which play a key role in stem cell maintenance, pluripotency, angiogenesis, survival, and self-renewal and repress cell differentiation.

The various physicochemical properties are very important to characterize the protein which was computed through the ProtParam Tool. The ProtParam Tool found that the molecular weight of Oct4 was about 38.57 KDa. The isoelectric point (pI) is the pH where the protein net charge will be zero but the surface will be enclosed with the charge which makes the

**Table 1. Tabulated representation of interacting network of Oct4 with other important proteins from STRING database.**

| Node 1 | Node 2 | Coexpression | Experimentally determined | STRING database annotated | Automated text-mining | Total score |
|---|---|---|---|---|---|---|
| POU5F1* | SALL4 | 0.162 | 0.465 | 0.9 | 0.882 | 0.994 |
| POU5F1 | SALL1 | 0.061 | 0.139 | 0.9 | 0.438 | 0.948 |
| POU5F1 | ZSCAN10 | 0.232 | 0.079 | 0.9 | 0.441 | 0.955 |
| POU5F1 | SMAD4 | 0 | 0.058 | 0.9 | 0.434 | 0.942 |
| POU5F1 | SOX17 | 0 | 0.475 | 0 | 0.9 | 0.945 |
| POU5F1 | SMAD2 | 0.049 | 0.064 | 0.9 | 0.555 | 0.955 |
| POU5F1 | ZIC3 | 0.097 | 0.152 | 0.9 | 0.504 | 0.957 |
| POU5F1 | STAT3 | 0.062 | 0.065 | 0.9 | 0.694 | 0.969 |
| POU5F1 | PRDM14 | 0.158 | 0.079 | 0.9 | 0.694 | 0.973 |
| POU5F1 | TDGF1 | 0.288 | 0.162 | 0.9 | 0.813 | 0.987 |
| POU5F1 | SOX2 | 0.089 | 0.538 | 0.9 | 0.984 | 0.999 |
| POU5F1 | KLF4 | 0.055 | 0.079 | 0.9 | 0.945 | 0.994 |
| POU5F1 | NANOG | 0.278 | 0.262 | 0.9 | 0.985 | 0.999 |
| POU5F1 | CDX2 | 0.062 | 0.266 | 0 | 0.927 | 0.945 |
| POU5F1 | EPAS1 | 0 | 0.262 | 0.9 | 0.326 | 0.945 |
| POU5F1 | FOXD3 | 0.116 | 0.379 | 0.9 | 0.744 | 0.984 |
| POU5F1 | FGF2 | 0 | 0 | 0.9 | 0.804 | 0.979 |
| POU5F1 | DPPA4 | 0.273 | 0.118 | 0.9 | 0.663 | 0.975 |
| POU5F1 | LIN28A | 0.356 | 0 | 0.9 | 0.943 | 0.996 |
| POU5F1 | PBX1 | 0 | 0.052 | 0.9 | 0.422 | 0.94 |

* POU5F1 generic name of Oct4.

protein stable and compact. The pI of the Oct4 was 5.69 indicating the acidic nature (pI<7) of the protein. The aliphatic index (AI) such as relative volume of the protein occupied by aliphatic side chains such as alanine, valine, isoleucine, and leucine considered as positive factors for the increase of thermal stability of globular proteins [78]. The AI value of Oct4 was 66.61 indicating that Oct4 protein may be stable for a wide range of temperatures. Smaller than 40 instability index considered as stable protein, above 40 considered that the protein may be unstable. The instability index of Oct4 is 53.24 which demonstrated that the protein is unstable. The GRAVY of Oct4 is -0.435 indicated the hydrophilic nature of the protein [79].

## Protein structure prediction and stability evolution

Since we hunt to identify small molecule inhibitors to abolish the activity of Oct4 thereby we focused on the Oct4 POU domain subdivided into POUs and POUHD domains (amino acid residues 138–287) [80] which is the most critical part of the protein for iPSC generation [12, 14]. MODELLERv9.19 was utilized to generate 50 3D coordinates of the Oct4 POU domain based on template PDB structure of *Mus Musculus* (PDB ID: 3L1P) with 88.67% identity and 100% query coverage (for residues 138–287) [81]. The model structure having a minimum DOPE score was considered as the best model and subjected for validation through ProSA, and SAVES server. The model structure was visualized and shown in Fig 3a and compared with the backbone structure of the template in Fig 3b. The accurate model must have the root mean square deviation (RMSD) value less than 2Å while the superimposed structure of the model protein and template was found to be RMSD of 0.102Å using Pymol [82] indicating both the proteins were well matched. PROCHECK examines the stereochemical quality of a protein structure by analyzing residue-by-residue geometry and overall structure geometry

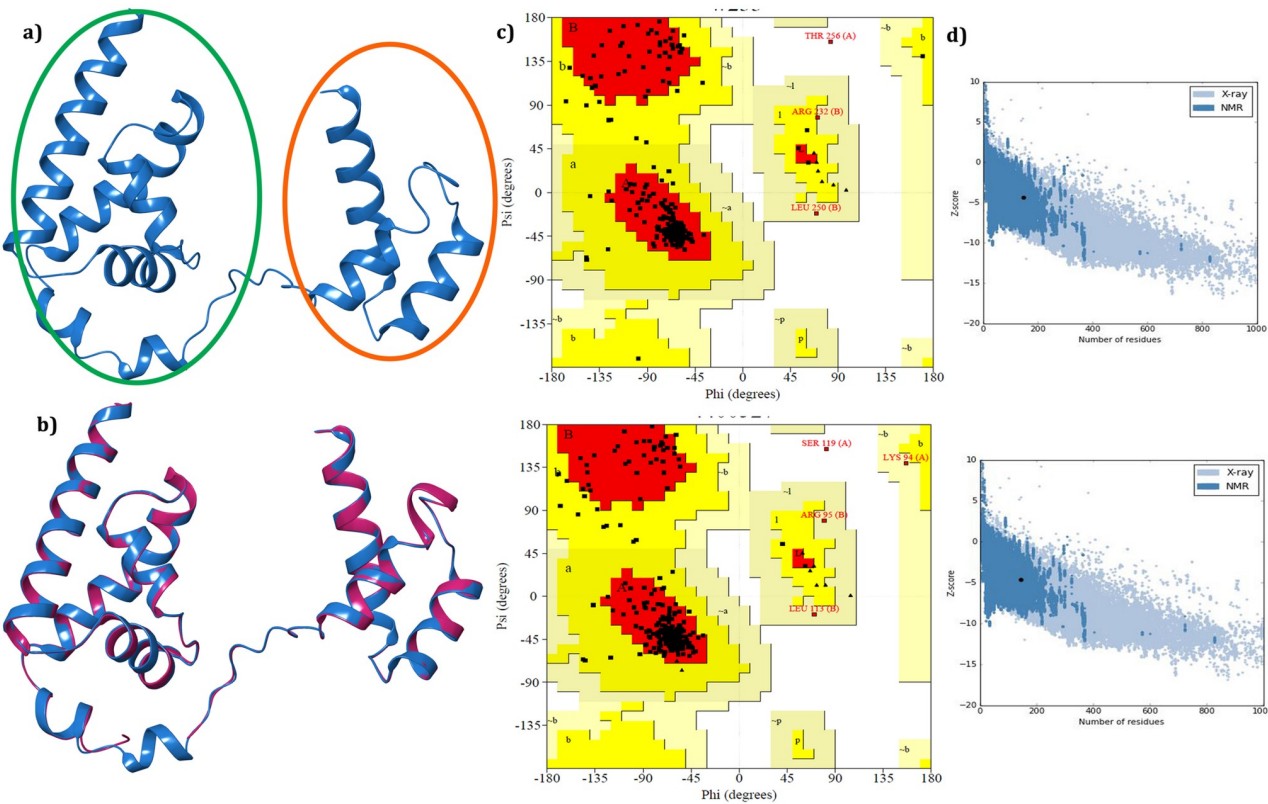

**Fig 3. Best 3D structure of the homology model Oct4 and validation result. a)** Model structure predicted through modeler where green color circle represented the POUs domain and red color showed POU$_H$ domain. **b)** Structural alignment of model structure with the crystal structure (PDB ID: 3L1P) where model structure displayed in blue color and crystal structure in deep pink color; **c)** Ramachandran plot for model structure (top) and template structure (3L1P) (bottom) showing the energetically allowed regions for backbone dihedral angles ψ against φ of amino acid residues; **d)** The calculated quality (Z) scores generated by ProSA in the context of all experimentally determined protein structures. The top panel Fig represents the modeled structure and the bottom panel represents the template structure (PDB ID: 3L1P).

and imparts amino acid residues distribution on the Ramachandran plot [83]. In comparison with template structure (3L1P) Ramachandran plot, it was evident that 90.2% of the residues from modeled Oct4 structure were in the most favoured region, 8.7% of the residues were in the additional allowed region, 0.7% of the residues were in the generously allowed region and 0.4% of the residues were in the disallowed region while for template structure (3L1P) 90.2% of the residues were in the most favoured region, 8.3% of the residues were in the additional allowed region, 1.1% of the residues were in the generously allowed region and 0.4% of the residues were in the disallowed region (Fig 3c). The model validation results from PROCHECK informed that the backbone dihedral angles of ψ and φ angles in the model were reasonably accurate. The secondary structure of the protein was predicted by PSIPRED [84] and provided in S1a Fig. ERRAT uses the statistical relation of non-bonded interactions between different atom types to analyze the structure. It evaluates the overall quality of the model and produces values around 95% or higher for high-resolution structures and values around 91% for low-resolution structures [85]. ERRAT predicted value of 92.53% for model structure and 94.66% for 3L1P suggested that the overall quality factor of the protein structure was satisfactory (S1b Fig). Furthermore, the predicted 3D coordinates of Oct4 was subjected to ProSA for checking the potential errors. ProSA calculates the Z-score which includes the overall quality of the model and deviation of total energy in the random conformation concerning to the energy

distribution of the predicted model [36]. The Z-score of the template 3L1P and target was
-4.68 and -4. 38 respectively indicating that the model structure was similar to the template
structure (Fig 3d). The knowledge-based energy for model structure and template structure
was displayed the S1c Fig. The results from various servers recommended that the predicted
model was more reliable in terms of main chain stereochemistry and overall quality factors.

Stability and fluctuations in the model protein Oct4 were evaluated by subjecting molecular
dynamics (MD) simulation using Desmond utility implemented in the Schrodinger's Maestro
suite. Root mean square deviation (RMSD), Root mean square fluctuation (RMSF), Radius of
gyration (ROG), and Secondary structure element (SSE) of Oct4 was monitored during the
100ns MD simulation. The backbone RMSD graph presented in Fig 4a explored that the
model structure was quite unstable throughout the simulation with an RMSD value from 6Å
to 17Å. The C-alpha RMSF supported RMSD results with an RMSF value of 3Å to 12Å and
inferred that the whole region of the model protein fluctuated with higher structural motion
(**Fig 4b**). ROG was analyzed to examine the compactness of the model protein which showed
that the structure was highly destabilized till 50ns with 18Å to 24Å leading to the loss of protein
compactness and the deviation was reduced after 60ns (Fig 4c). SSE was investigated to check
overall protein structure stability which confirmed that the protein structure had average con-
formation with 60% SSE, mainly composed of helices and loops rather than strands and turns
that showed conformational changes during MD simulation (Fig 4d). As per the report from
**Yesudhas et al.**, Oct4 displayed more deviation due to only separation of 3 base pairs from the

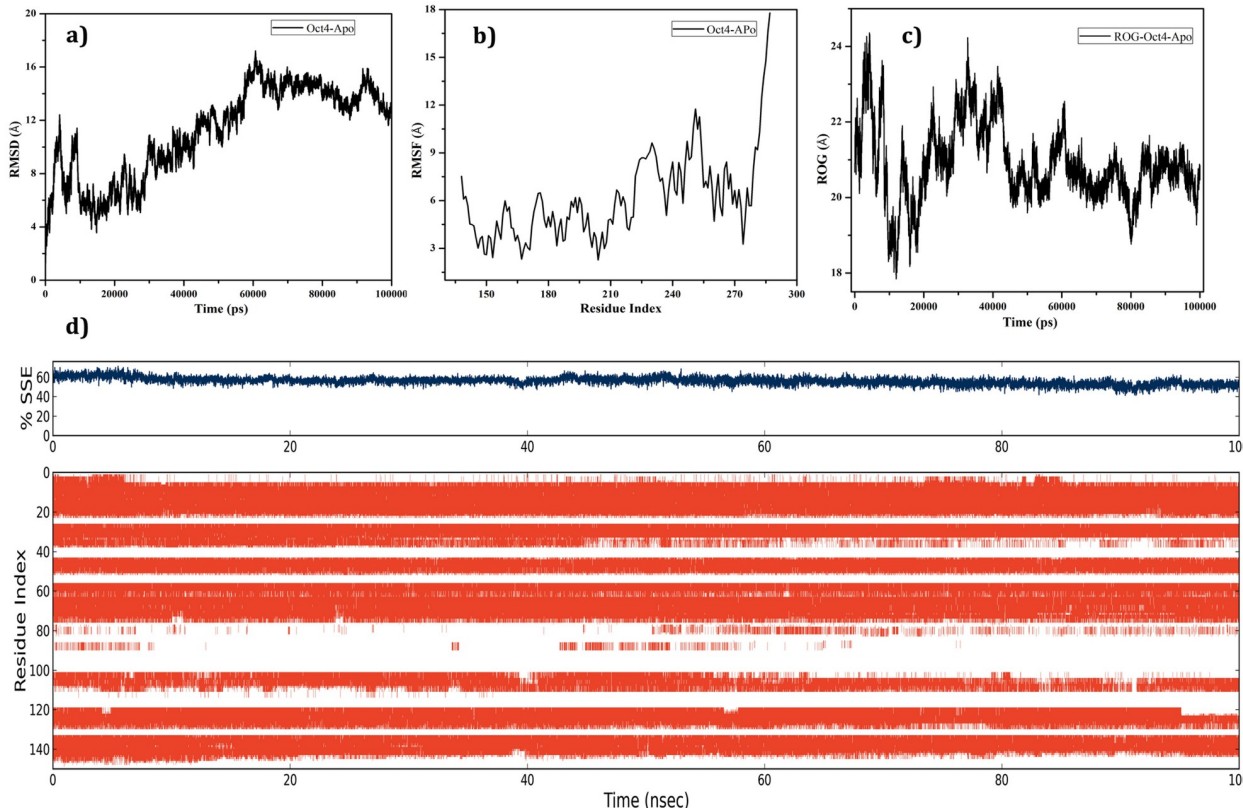

**Fig 4. Stability evaluation results of predicted 3D coordination of Oct4 through molecular dynamics simulation. a)** Backbone RMSD; **b)** C-
alpha RMSF of Oct4; **c)** ROG for Oct4; **d)** SSE percentage and its occupancy of helices (orange), strands, turns and loops (white) over 100ns time
period with reference to the index of the residue.

Oct4 and Sox2 DNA-binding sites [86] then it is obvious to exhibit high deviation and instability of Oct4 without any interacting proteins or DNA. The dynamics results suggested that the system showed a noticeable aspect of conformational changes which may be stable in the presence of some macromolecules or small molecules.

## Druggable pocket prediction, ligand filtration, and identification of potent inhibitors through virtual screening

The final validated model was used as an input structure to identify potentially druggable pocket for virtual screening using the SiteMap tool of Schrodinger. SiteMap predicted three druggability sites i.e. Site-1, -2, and -3 with site scores 0.730, 0.718, and 0.614 respectively. Details of other outputted results from this calculation presented in Table 2. The site score was calculated by analyzing various physical descriptors such as size, degree of enclosure, degree of exposure, tightness, hydrophobic, hydrophilic, hydrogen bonding possibilities, and linking site points that contribute to the binding of ligand with protein [87]. By considering site score, druggability score (Dscore), and volume, Site-1 was considered for further screening of potent small molecules and displayed in Fig 5a. The site score and Dscore of one or above for each considered a suitable site for the ligand binding and regulating the activity of protein with compounds [87]. The selected site having scores near to one was thought to be an appropriate active pocket for binding of the drug-like compounds. Further, the accuracy of the predicted binding site was evaluated by superimposing with the template structure 3L1P (complex with DNA) which showed that the selected druggability pocket (Site-1) was well placed at the binding site of DNA in the crystal structure (Fig 5b). Site-1 was found on the $POU_{HD}$ domain of Oct4 which interacts with the DNA sequences. The superimposition of DNA and modeled Oct4 protein labeled with active site residues has been displayed in Fig 5c. Some studies verified through crystal structure [88] and simulation [89] that $POU_{HD}$ Domain bound to nonconsensus DNA sequences which is a key in the biological function of Oct4 including stem cell maintenance, self-renewal, and pluripotency [90, 91]. The validated results indorsed that druggability Site-1 was the best site to generate a receptor grid for further molecular docking studies.

In this study, optimized natural compound databases ZincNPD, NCINPD, and NPB containing a total of 155819, 55958, and 8395 compounds (with their conformers) were subjected to ADME filtration to discriminate the drug-like compounds targeting the central nervous system. This process provided the advantages to reduce false positives and elude compounds having poor pharmacokinetic outlines [92]. As mentioned in the materials and method section, the significant ADME parameters were calculated using QikProp [52] and a standalone library was formed compressing a total of 101572 compounds as results of the defined filtration criteria. Further, the filtered compounds were carried forward for receptor-based virtual screening (RBVS) to identify potent small molecule inhibitors against Oct4. Before performing RBVS, the receptor grid was generated by prearranging a selected sitemap (Site 1) on the protein active site to calculate the potential docking score of the drug-like compounds. A multi-tried screening protocol was performed starting from high throughput virtual screening (HTVS),

**Table 2. SiteMap generated druggability site score results.**

| Number of Sites | Site Score | Dscore | Volume | Number of Amino acid residues |
|---|---|---|---|---|
| Site 1 | 0.730 | 0.682 | 134.799 | 234,235,246,237,280,281,273,284,242, 276,277 |
| Site 2 | 0.718 | 0.652 | 68.257 | 146,139,190,192,193,194,142,197,143, 198 |
| Site 3 | 0.614 | 0.497 | 64.141 | 169,224,225,227,228,229,172,173,178 |

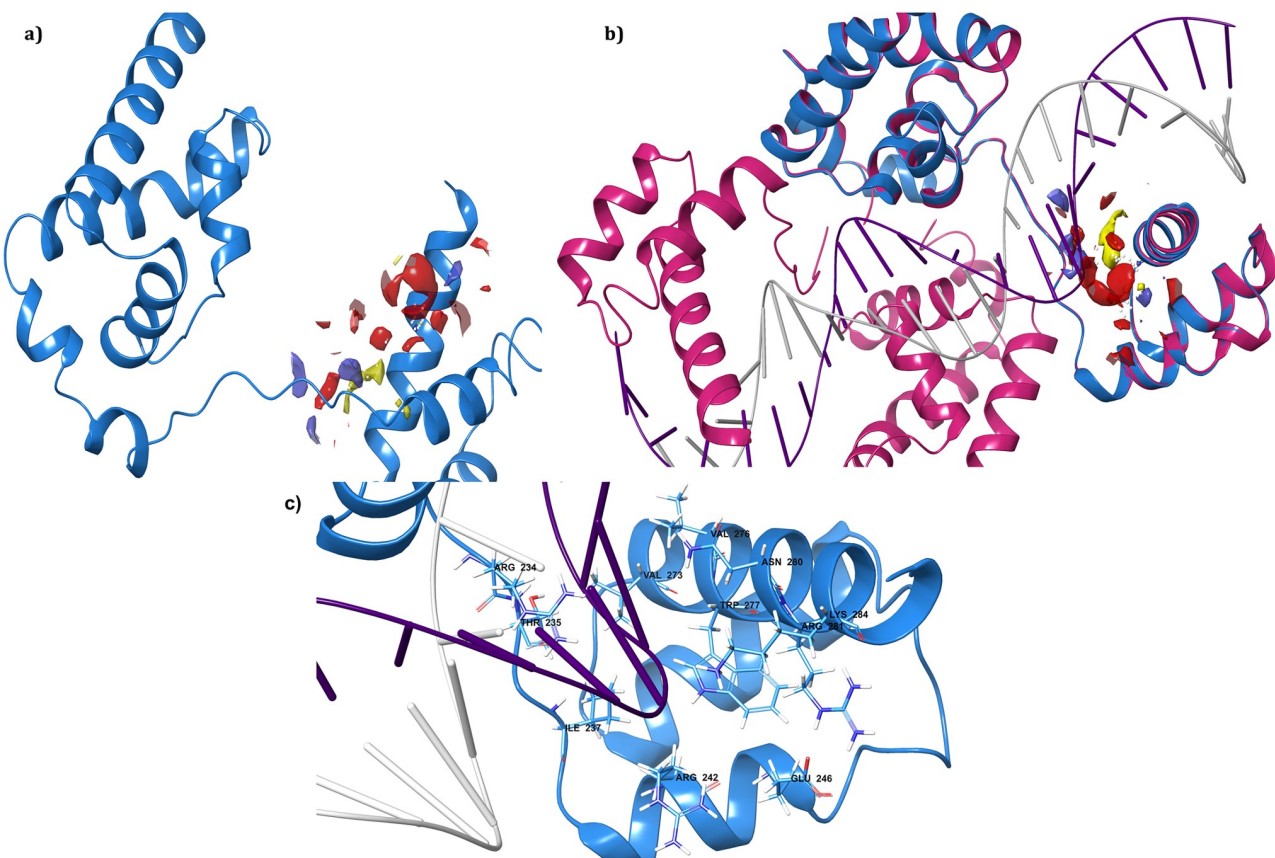

**Fig 5. The identified druggability region of Oct4 by SiteMap. a)** the active region of Oct4 with the different colored surface where red represents negative, blue represents positive and yellow represents hydrophobic regions; **b)** Superimposition of predicted druggability pocket with the co-crystal DNA complex protein (PDB ID: 3L1P) where model structure displayed in blue color and crystal structure in deep pink color; **c)** close-up representation of binding site with amino acid name and numbers.

followed by standard precision (SP) to extra precision (XP) where ligand was considered as flexible and protein as rigid [59, 93]. HTVS screens large databases very fast by reducing the number of intermediate conformations with rough scoring function while SP is ten times slower and more accurate than HTVS. XP performs extensive sampling by employing a more sophisticated scoring function than the SP scoring function. XP specifically eliminates false positives that collected through SP by penalizing the ligands based on the ligand-receptor shape complementarity [48, 54]. Top 10157 (10% of 101572) compounds were considered from the HTVS docking and subjected to SP docking. Among 10157 compounds, 1015 compounds from the SP docking were passed through XP docking. Lastly, we got 101 complexes as a result of XP docking. The top 30 complexes were examined through visualization in ordered to better understand the binding mechanism of the compounds (**S1 Table in** S1 File). Based on docking score and interaction study 13 compounds were chosen and subjected to induce fit docking for further estimation of binding mode. The predicted drug-like properties of the selected compounds were listed in **S2 Table in** S1 File. All the selected compounds displayed good pharmacokinetic properties under the satisfactory range including CNS activity, blood-brain barrier permeability, and human oral absorption which appeared to be suitable as leads for further development of anti-GBM drugs.

**Table 3. Induce fit docking results of top thirteen screened potential compounds with different score and interactions with protein.**

| Sl. No. | Compound ID | IFD Score | Docking Score | Glide energy | Glide e-model | H-bond interacting residues | Other interacting residues |
|---|---|---|---|---|---|---|---|
| 1. | ZINC20410920 | -338.45 | -9.895 | -55.504 | -57.821 | R242, I237, S236, N280, R234 | *Sltb: K284 |
| 2. | ZINC02145000 | -333.34 | -9.441 | -52.716 | -67.745 | T235, R242, N280 | *Sltb: K284 |
| | | | | | | | Pi-pi-W277 |
| 3. | NSC292567 | -331.08 | -8.389 | -42.882 | -53.854 | R232, R234, W277, N280 | *Saltb: K231 |
| 4. | NPB7083 | -329.30 | -7.391 | -61.011 | -90.673 | R281, N280, W277, R234, K284, R242 | Pi-pi: W277 |
| | | | | | | | #Pi-cat: K284, R234 |
| 5. | ZINC00519647 | -329.10 | -7.687 | -43.550 | -49.652 | R281, N280, W277, R234, T235, R242 | |
| 6. | ZINC85876856 | -328.87 | -6.099 | -45.740 | -63.145 | W277, K284, R234, T235 | #Pi-cat; R242, R234 |
| 7. | ZINC08764609 | -328.40 | -8.775 | -48.687 | -56.841 | N280, K284, R234, R242 | |
| 8. | ZINC15967742 | -328.27 | -7.493 | -56.415 | -67.500 | R234, T235, R242, K285 | *Sltb: R232 |
| | | | | | | | Halogen: K233 |
| | | | | | | | #Pi-cat: R242 |
| 9. | ZINC32124203 | -328.13 | -7.164 | -45.871 | -60.598 | K284, N280, W277, R242 | *Sltb: R234 |
| | | | | | | | #Pi-cat:K284 |
| 10. | ZINC02092319 | -326.68 | -6.862 | -57.754 | -87.322 | K284, N280, T235 | #Pi-cat: K233 |
| 11. | ZINC14759216 | -326.18 | -6.964 | -46.202 | -67.032 | E246, N280, T235 | Pi-pi: W277 |
| 12. | NSC83439 | -325.06 | -8.359 | -33.945 | -46.796 | R242, K284, T235, N280 | *Sltb: K284 |
| 13. | NPB4533 | -324.95 | -6.466 | -47.752 | -66.822 | K284, N280, R234, I237 | |

* Sltb: Salt Bridge.

# Pi-cat: Pi-cation.

## Lead conformation validation through induce fit docking (IFD) and pose rescoring with MMGBSA

The thirteen selected compounds were redocked through induce fit docking to evaluate the binding mode. The most important feature of IFD is that the ligand, active site residues of the protein and its vicinity are considered as flexible for better assessment of protein-ligand interactions [62, 94]. The IFD results of 13 compounds were given in Table 3 and shown in Fig 6a which clarified that the screened compounds had more affinity to Oct4 with docking score from -6.099 to -9.895Kcal/mol. The docking results demonstrated that ZINC20410920 exhibited the highest IFD score (-338.45Kcal/mol) and docking score (-9.895Kcal/mol) with the development of five hydrogen bonds with R242, I237, S236, N280, and R234 and one salt bridge with K284. ZINC02145000 was the second-highest scored hit with an IFD score of -333.34Kcal/mol which communicated with Oct4 by establishing three hydrogen bonds with T235, R242, and N280; one salt bridge with K284; and one pi-pi interaction with W277. The compounds NPB7083 and ZINC00519647 interacted with protein by forming six hydrogen bonds with IFD scores of -329.30Kcal/mol and -329.10Kcal/mol respectively. Both NPB7083 and ZINC00519647 established nearly similar interactions with less differentiation, for instance, four common hydrogen bonds with R281, N280, W277, R234, and R242. The variation was that NPB7083 showed one hydrogen bond interaction with K284; two pi-cat interactions with K284, and R234; and one pi-pi connection with W277 whereas ZINC00519647 showed only one hydrogen bond with T235. The amino acid residues R232, R234, W277, and N280 of the protein involved in hydrogen bond interactions and K231 participated in salt bridge formation with compound NSC292567. Mostly, all the selected hits showed common interactions with the amino acid residues R281, N280, W277, R234, T235 and R242 which showed similar interactions as stated by literature that the amino acid residues R234, R242,

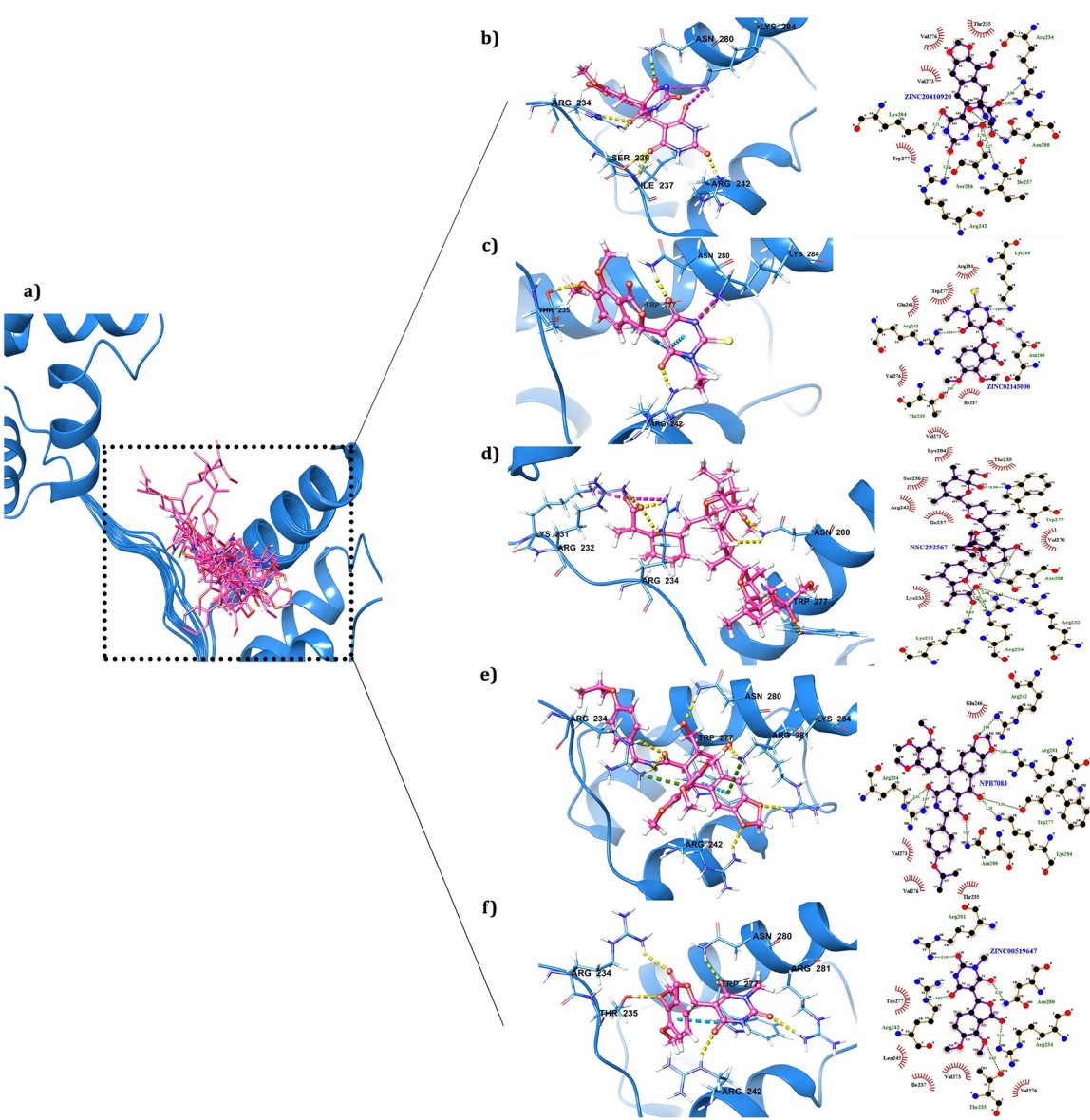

**Fig 6. A close view of interacting residues of Oct4 establishing connections with screened compounds. a)** A glance of thirteen screened compounds binding mode inside the Oct4 DNA binding pocket; **b)** the 3D image of ZINC20410920 displayed its communication with Oct4 by forming H-bonds and 2D interaction map generated by LigPlot representing the hydrophobic interaction; **c)** ZINC02145000 interacting image; d) NSC292567; **e)** NPB7083; **f)** ZINC00519647.

K254, R275, N280 and Q283 from POUHD domain were establishing consistent interactions with DNA [13, 86]. The 3D interaction of the best five compounds of IFD docking was illustrated in Fig 6b-6f where hydrophobic interactions were presented in LigPlot schematic diagrams [95]and other compound interactions displayed in S2 Fig. The XP docking followed by IFD docking results concluded that a funnel base docking can provide effective compounds [46].

In the comparison of IFD with XP docking scores and interactions, it was evidenced that ZINC20410920 and ZINC02145000 maintained the same interactions and docking scores as XP docking (Table 3 and **S1 Table in** S1 File). All compounds showed variable IFD score and

**Table 4. Calculated binding free energies (Kcal/Mol) of the screened compounds.**

| S. No. | Compound ID | ΔG Bind | ΔG Bind coulomb | ΔG Bind vdw | ΔG Bind covalent | ΔG Bind solv GB | ΔG Bind Lipo | ΔG Bind Hbond |
|---|---|---|---|---|---|---|---|---|
| 1. | NPB7083 | -85.28 | -42.64 | -51.76 | 3.34 | 45.19 | -29.99 | -4.72 |
| 2. | ZINC02092319 | -81.91 | -30.95 | -49.58 | 1.50 | 27.82 | -24.34 | -2.58 |
| 3. | ZINC15967742 | -66.16 | -188.44 | -40.03 | 4.11 | 180.02 | -12.98 | -6.64 |
| 4. | ZINC20410920 | -82.80 | -385.72 | -37.16 | 2.01 | 358.51 | -13.62 | -6.60 |
| 5. | ZINC02145000 | -59.46 | -477.75 | -31.57 | 1.31 | 465.59 | -12.08 | -3.28 |
| 6. | ZINC08764609 | -71.18 | -176.91 | -36.39 | 1.29 | 170.70 | -23.72 | -4.31 |
| 7. | NPB4533 | -59.72 | -32.02 | -40.74 | 2.11 | 31.48 | -17.80 | -2.72 |
| 8. | ZINC14759216 | -66.36 | -31.36 | -33.19 | 1.05 | 25.33 | -20.06 | -2.99 |
| 9. | ZINC32124203 | -59.26 | -221.64 | -36.54 | 1.28 | 218.43 | -12.18 | -4.19 |
| 10. | ZINC85876856 | -60.03 | -35.88 | -31.31 | -0.22 | 31.26 | -13.48 | -4.15 |
| 11. | ZINC00519647 | -55.79 | -231.64 | -34.05 | 5.04 | 225.82 | -15.25 | -2.97 |
| 12. | NSC292567 | -70.97 | -208.81 | -37.47 | 2.52 | 193.07 | -13.17 | -7.11 |
| 13. | NSC83439 | -51.52 | -145.10 | -28.66 | 3.47 | 137.66 | -15.37 | -3.52 |

docking energy in comparison to XP docking but when we considered interaction wise, we found that the selected compounds established almost the same interactions as XP docking along with some extra connections. For instance, NSC292567 developed interactions with R232, R234, W277, N280, and K231 in IFD docking whereas in XP docking it showed its engagement with T235, R234, and R232 residues only. The compound NPB7083 displayed its connection with R281, N280, W277, R234, K284, and R242 in IFD docking however in XP docking it communicated with only amino acid residues N280, K284, T235, and R234 of Oct4.

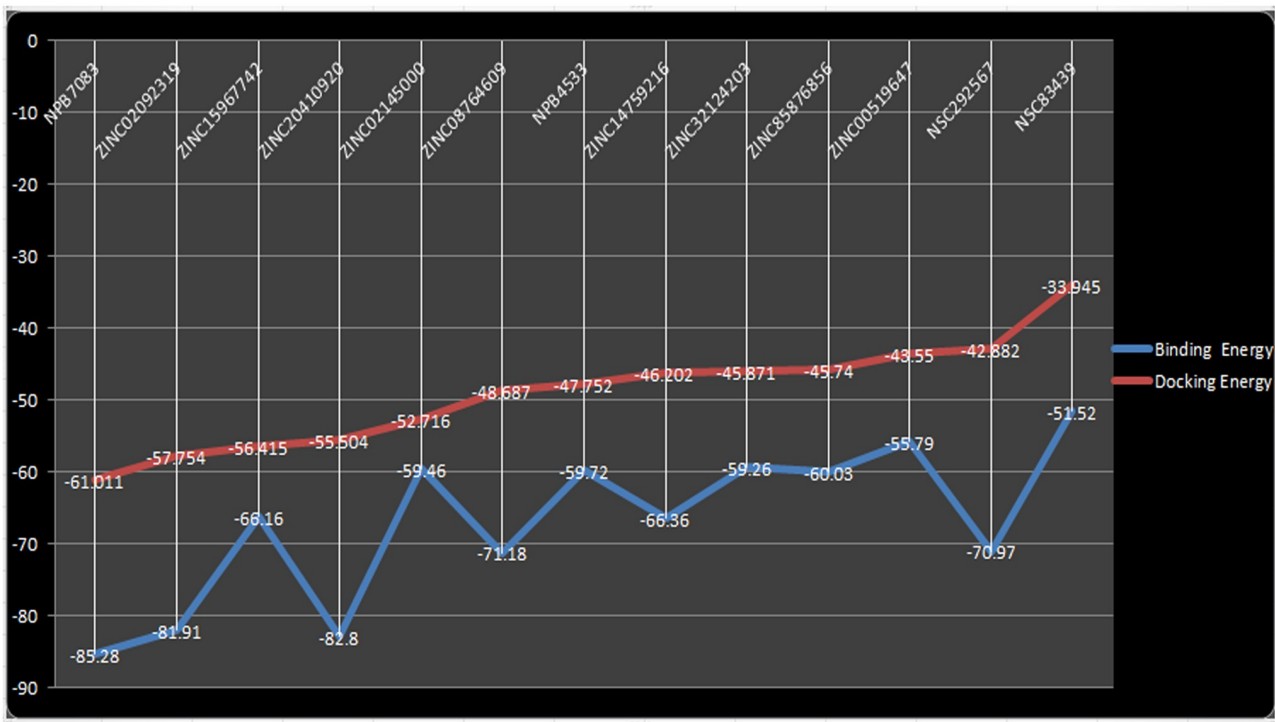

**Fig 7. Docking energy and binding energy comparison of screened compounds.**

ZINC00519647 exhibited six interactions in IFD docking with the amino acid residues R281, N280, W277, R234, T235, and R242 and it showed connections with the residues K284, N280, T235, and R234 in XP docking. Based on both rigid and flexible receptor with better scoring and interacting amino acid residues, the screened hits were taken for further validation through binding free energy calculation and molecular dynamics simulation.

The confirmations of the docking results and ligand efficiency were determined by calculating binding free energy through prime-MM/GBSA (Molecular Mechanics/Generalized Born Surface Area) [60]. Additionally, the MM-GBSA scoring usually provides a significant correlation with experimentally determined data [96, 97]. The calculated binding free energy for the screened compounds using the OPLS-AA force field and the GBSA continuum solvent were represented in Table 4. The calculated binding free energy of the different screened hits was ranged from -51.52 to -85.28Kcal. The ligand molecule was considered as best which utilizes less energy to comfort inside the active pocket of the protein [94]. Docking energy and binding energy were compared to validate the docking pose and the comparison result was displayed in Fig 7. As predicted by MMGBSA, NPB7083 exhibited the highest negative binding free energy (-85.28Kcal/mol) which correlated well with the docking energy (-61.011Kcal/mol). The compound ZINC20410920 was predicted as the second-best binder by the MM/GBSA calculation with a calculated binding free energy of -82.80Kcal/mol and docking energy of -55.504Kacl/mol. some results of MM/GBSA imparted dissimilarity with the docking energy, for instance, NSC292567 was established to possess less docking energy (-42.882Kcal/mol) but showed good binding free energy (-70.97Kcal/mol). Another example of the variation between MM/GBSA and docking energy was that the compound ZINC02145000 exhibited good docking energy of -52.716Kcal/mol however displayed low binding free energy of -59.46Kcal/mol. The compound NSC83439 hypothesized to have the poorest binding free energy (-51.52Kcal/mol) than all selected compounds which were also matched its docking energy (-33.945Kcal/mol). In summary, MM/GBSA calculation classified the binding affinity of the top highest and lowest compounds in the exact place as docking energy while there were little changes in the order for intermediate docking energy holding compounds. Thereby, we have taken all the selected complexes for the further validation of the compound stability through molecular dynamics simulation.

## Interaction and stability validation of selected leads through molecular dynamics simulation

To investigate the effective conformation of hits, the stability of the protein and ligands, RMSD was examined for each complex with respect to the initial structure through molecular dynamics simulation for 50ns. RMSD determined the stability of the protein comparative to its conformation by analyzing the deviation produced during the simulation. The smaller RMSD value signifies the more stable structure of the protein. The average RMSD of each complex and free model of Oct4 was calculated after 50ns and plotted in Fig 8a. The backbone RMSD plot of selected complexes concerning their initial structures indicated that most of the complexes showed high deviation throughout the simulation which showed the active form of the ligands inside the binding pocket. Four complexes such as NPB7083, NSC292567, ZINC08764609, and ZINC14759216 showed major differences in RMSD with average RMSD value of 12.85 Å, 15.2 Å, 12.08 Å, and 11.54 Å respectively when compared with the free form of Oct4 (average RMSD of 11.11Å). Further, NPB4533, NSC83439, ZINC02145000, and ZINC32124203 complexes exhibited the least deviation with average RMSD values of 6.17Å, 6.489, 5.7Å, and 6.1Å respectively suggesting that these complexes were more stable than other complexes (separately depicted in S3a Fig). From these four least RMSD complexes, the

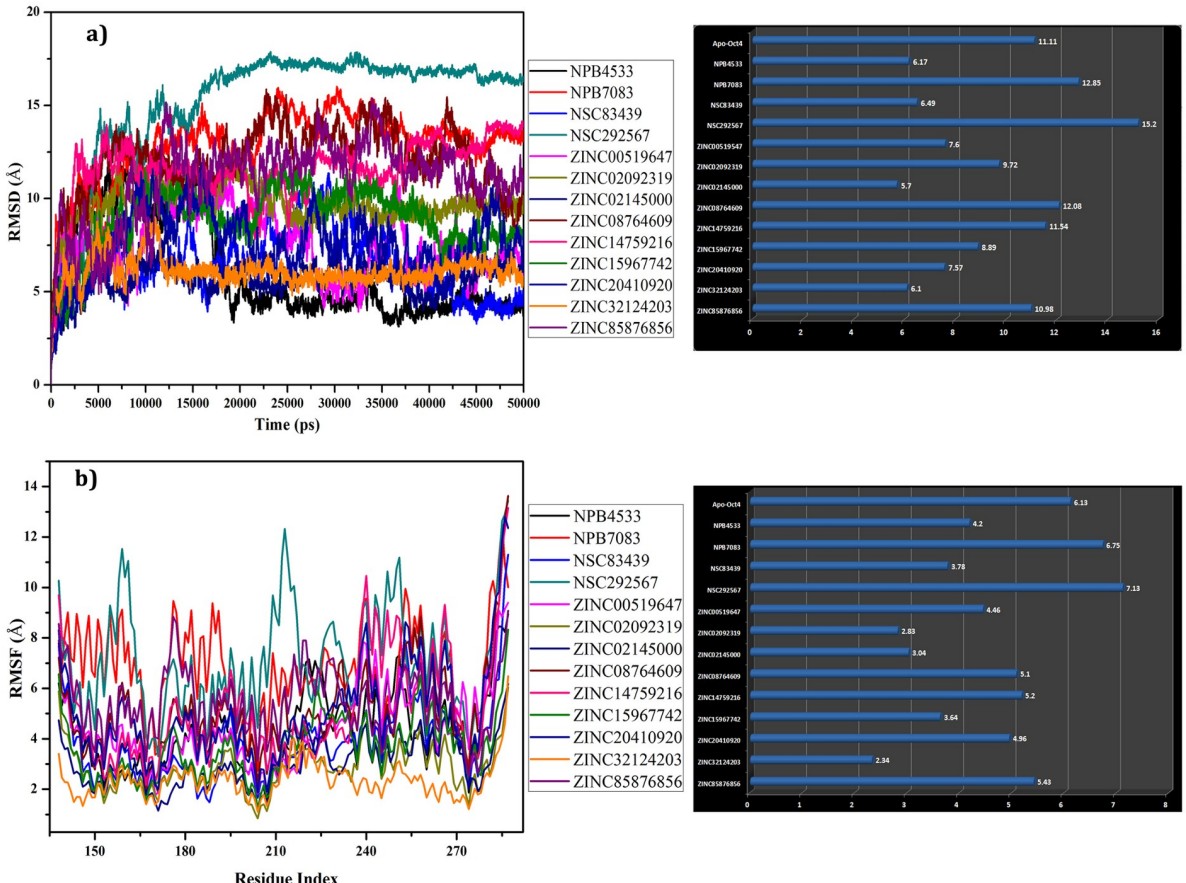

**Fig 8. Molecular dynamics simulation results over 50ns. a)** Time dependent RMSD (Å) of backbone atoms of protein displayed with average mean variation **b)** C-alpha RMSF of various mobile regions of the complexes with respect to the residues and mean variation plot of average RMSF values.

complex NPB4533 remained unstable in the first 16ns of the simulation and after 16ns sustained stability till the end of 50ns simulation with RMSD of 4.5 Å. Although the Oct4-NSC83439 complex displayed the least average RMSD value, it was characterized by a higher RMSD deviation during the simulation. The complex ZINC02145000 persisted in steady form during the entire simulation run with RMSD value ranging from 5Å to 6Å. From the starting of the simulation (1ns), the ZINC32124203 complex exhibited a higher RMSD of ~7.5Å due to the presence of a linker region between POUs and POUHD domains. The plateau was observed after approximately 11ns for ZINC32124203 and continued until the end of the simulation.

The residual fluctuation of all the complexes was analyzed by C-alpha motion and local changes in the secondary structure elements to monitor the various flexible region involving in the ligand binding (Fig 8b). The C-alpha Root mean square fluctuations (RMSFs) of all complexes revealed that most of the complexes experienced high fluctuation and supported the RMSD results. The mean variation plot was plotted to examine the average RMSF value of the compounds (Fig 8b). Results from the average RMSF analysis unveiled that except NPB7083 and NSC29567 complexes, all other complexes exhibited less average RMSF value than the free form of Oct4. The RMSF result of four selected complexes from RMSD analysis was examined separately which showed maximum fluctuation at the linker region (amino acid

residues 213–229) (S3b Fig). Further, the ligand-bound protein experienced a low fluctuation in comparison to free Oct4 protein despite the presence of the linker region (amino acid residues 213–229) which enabled the compounds to establish strong interaction with the protein during the whole simulation period. The POUs domains of the ZINC32124203 and ZINC02145000 complexes also displayed the least variation which was mainly due to the interaction of compounds with this region.

The SSE was monitored and analyzed to investigate the overall protein structure stability after ligand binding throughout the simulation (S4 Fig). Results from SSE demonstrated that all the structures maintained the structural conformation with an approximate SSE of 60% constituting helices rather than strand or loops, with some exceptions of NPB7083, ZINC20410920 and ZINC08764609. Table 5 displayed the ratio of SSE in initial structure (apo Oct4) and its complexes which suggested that all the complexes possessed stable conformation during the simulation. The SSE results also confirmed that the structure of the selected four complexes maintained the stable conformation during the entire time of simulation shown in Fig 9. Additionally, the linker region (amino acid residues 76–92) of four complexes showed stable conformational changes like from loop to alpha-helix formation throughout the simulation whereas the free model did not display any significant changes (Fig 9). Interestingly in the complex NPB4533, it was observed that the loop region at the position 36–40 and 95–98 converted to strands at 22ns and continued till 27ns which indicated that binding of the ligand possibly affecting the overall stability and conformational status of the structure.

The 50ns long simulation trajectories were inspected to evaluate the stability of the hydrogen bonds (H-bond) formed by screened compounds with Oct4 at the ligand-binding site (Fig 10a). The total number of H-bonds were calculated for all 13 complexes and demonstrated in Fig 10b. More interestingly, the results exposed that most of the compounds formed H-bond with W277 of Oct4 and occupied more than 90% of the simulation time (Fig 10c-10f and S5 Fig). There were a continuous disappearance and reappearance of the H-bonds found in between screened compounds connecting with different amino acid residues of Oct4 during the time of simulation in comparison to docking interaction. For instance, docking of the compound ZINC20410920 showed interactions with R242, I237, S236, N280, and R234 while in simulation W277 and K284 exhibited greater occupancy (S5 Fig). This may be to adopt a favorable ligand binding site due to the initial docking conformational changes during the

**Table 5. Showing the ratio of secondary structure elements distributed by the protein throughout the simulation.**

| Sl. No. | Protein/complex name | %helix | %strand | %total SSE |
|---------|----------------------|--------|---------|------------|
| 1. | Apo Oct4 | 56.24 | 0.00 | 56.24 |
| 2. | NPB4533 | 59.31 | 0.08 | 59.40 |
| 3. | NPB7083 | 53.78 | 0.00 | 53.78 |
| 4. | NPB83439 | 60.71 | 0.00 | 60.71 |
| 5. | NSC292567 | 58.09 | 0.00 | 58.09 |
| 6. | ZINC00519647 | 57.48 | 0.00 | 57.48 |
| 7. | ZINC02092319 | 59.01 | 0.00 | 59.01 |
| 8. | ZINC02145000 | 58.29 | 0.00 | 58.29 |
| 9. | ZINC08764609 | 53.23 | 0.06 | 53.29 |
| 10. | ZINC14759216 | 56.58 | 0.00 | 56.58 |
| 11. | ZINC15967742 | 57.62 | 0.00 | 57.62 |
| 12. | ZINC20410920 | 52.63 | 0.00 | 52.63 |
| 13. | ZINC32124203 | 57.84 | 0.00 | 57.84 |
| 14. | ZINC85876856 | 59.31 | 0.00 | 59.31 |

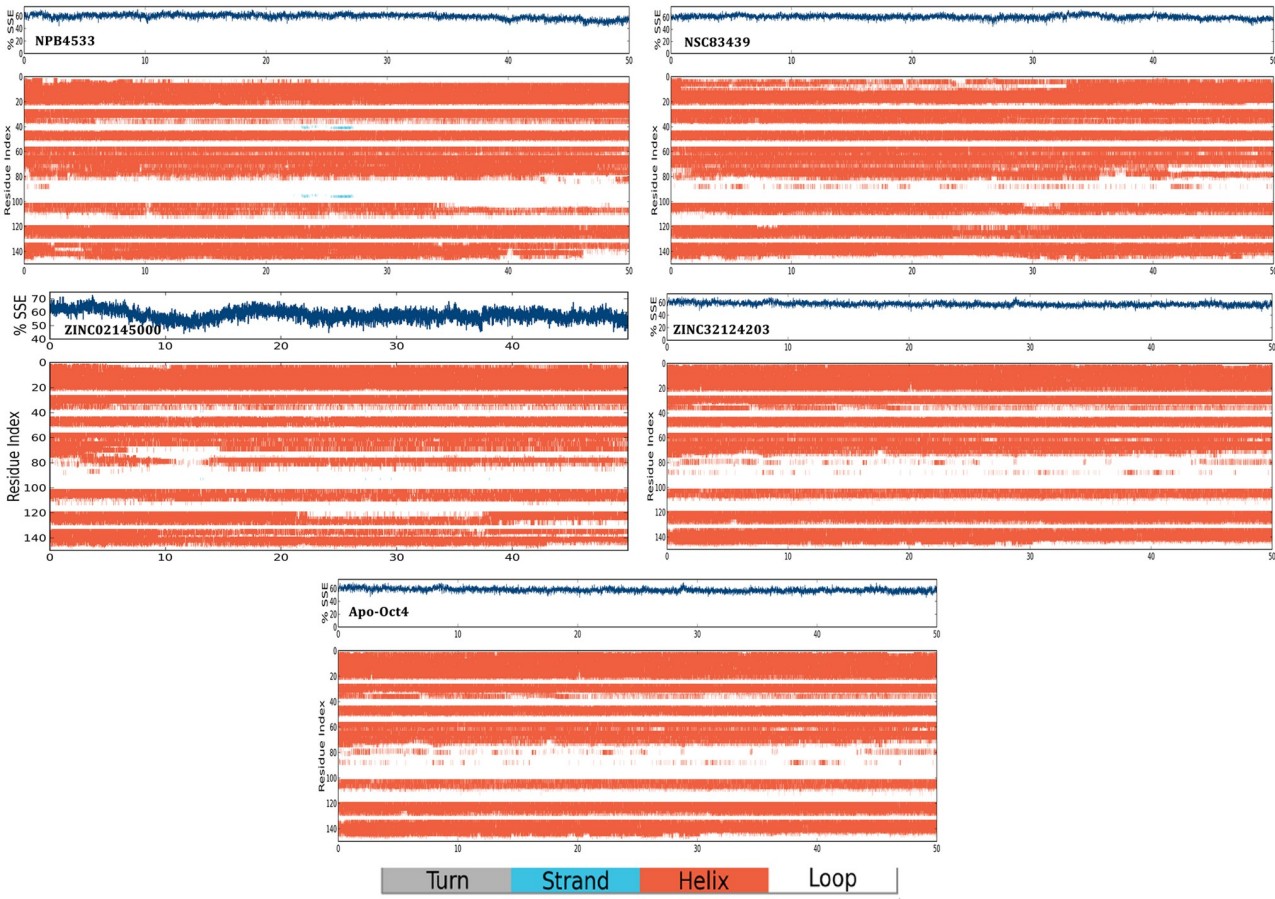

**Fig 9. Time evaluation of secondary structure elements for model structure and in complex with four selected compounds.**

time of the simulation. Further, a special investigation of selected four complexes revealed that the compounds ZINC02145000 and ZINC32124203 sustained stable and strong interactions with Oct4 during the simulation. ZINC02145000 compound presented similar kind of interactions with the T235, R242, N280, K284 (salt bridge), and W277 (Pi-pi) as found in the initial docking conformation, with some exception such as the interaction of R280 shifted to R281 and extra involvement of K284, and W277 in hydrogen bond formation with high occupancy (Fig 10d). ZINC32124203 changed its communication with Oct4 from N280, W277, R242, R234 (salt bridge) and K284 (Pi-Cation) in the initial docking conformation to I237 and V273 with the additional connection of R234 and K284 with 70–80% occupancy during the simulation (Fig 10f). The compound NSC83439 shifted its H-bond interactions from R242, K284, T235, and N280 from initial docking conformation to R232, L233, R287, and I237 during the simulation time (Fig 10c). The H-bond interactions were less prominent for compound NPB4533 while it formed hydrophobic interactions instead of H-bonds to remain inside the binding pocket of Oct4 (Fig 10e). Other than H-bonds, all the compounds engaged the protein by the hydrophobic, salt bridge and water-mediated communications. Altogether, H-bond analysis explored that all compounds established strong and stable connections with the POU$_{HD}$ domain. From H-bond analysis, we were surprised that despite strong and stable interactions formed by all the compounds with the DNA binding residues of the POU$_{HD}$

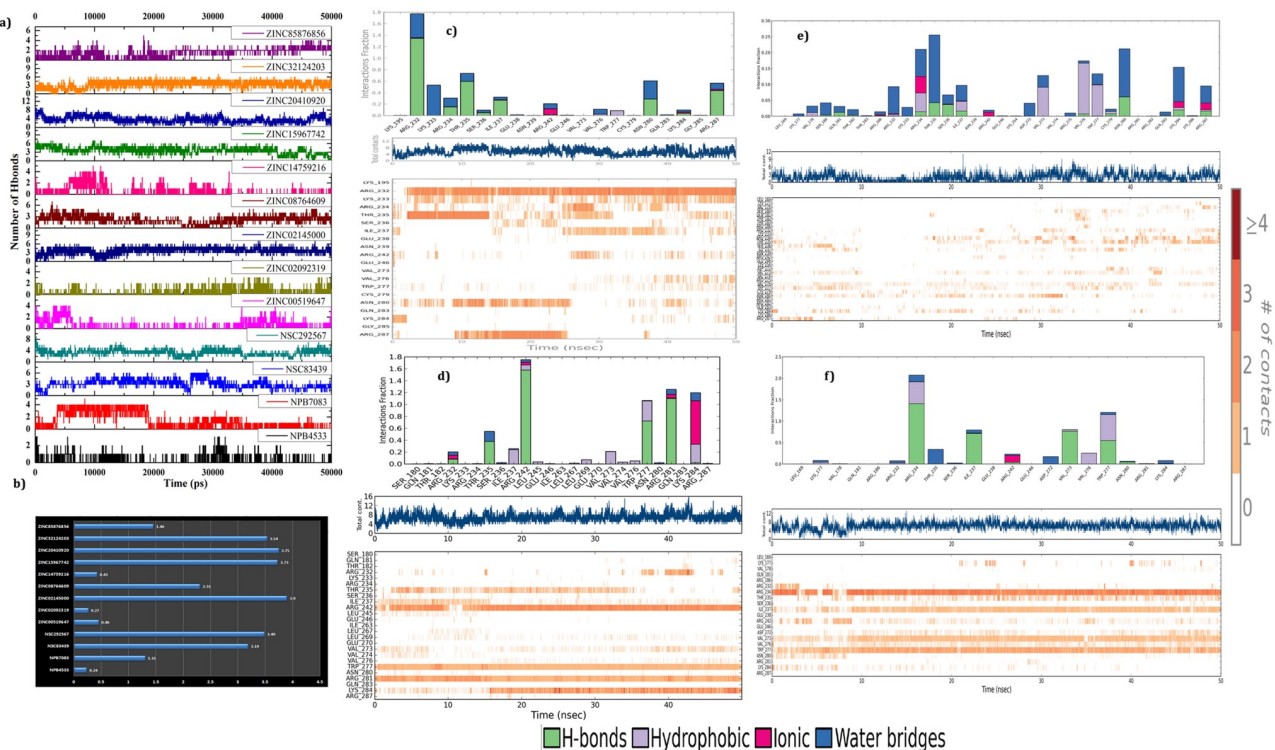

**Fig 10. The intermolecular hydrogen bonds formed by screened compounds with Oct4 during the 50ns molecular dynamics simulation. a)** Number of H-bonds established between protein-ligand complexes; **b)** Average mean variations plot of H-bond interaction; **c)** The H-bond occupancy throughout the trajectory for NSC83439, **(d)** ZINC02145000, **(e)** NPB4533 and **(f)** ZINC32124203 where the top panel represents the overall interaction of the protein with ligands and bottom panel unveiled the name of interactive amino acid residues.

domain, the backbone and C-alpha of Oct4 were showing high deviation during 50ns long simulation.

Therefore, we compared the simulation results with the docking energy and biding free energy we found that NPB7083 and ZINC20410920 having higher scores but showing high deviation whereas ZINC32124203 and ZINC02145000 possessing medium score exhibited stable conformation. For the further confirmation of the atomic movement and their collective or correlated residual motions in the complexes, principal component analysis (PCA) in combination with cross-correlation analysis were executed to obtain insight into the protein domain movement in four complexes and compared with apoprotein. PCA, a powerful tool that has been widely used to probe experimentally detected conformational variations [98]. PCA was applied to the backbone atoms in free Oct4 and four complexes. The principal motion of protein can be visualized and interpreted as porcupine plots by representing the eigenvectors that show the direction and magnitude of each of the backbone atoms [99]. The complete PCA analysis and the contributions of the different regions from the free Oct4 and four complexes were illustrated in Fig 11. The direction of arrows attached to the backbone atom of proteins signified the specific direction of motion, while the length of the arrows characterized the strength of the movement and also defined the essential conformational variation exhibited by the protein. The obtained plot illustrated that free protein inhabited comparatively more prominent motions and showed higher fluctuations in contrast to the compound bound proteins suggesting the major conformational variation occurred in apoprotein during

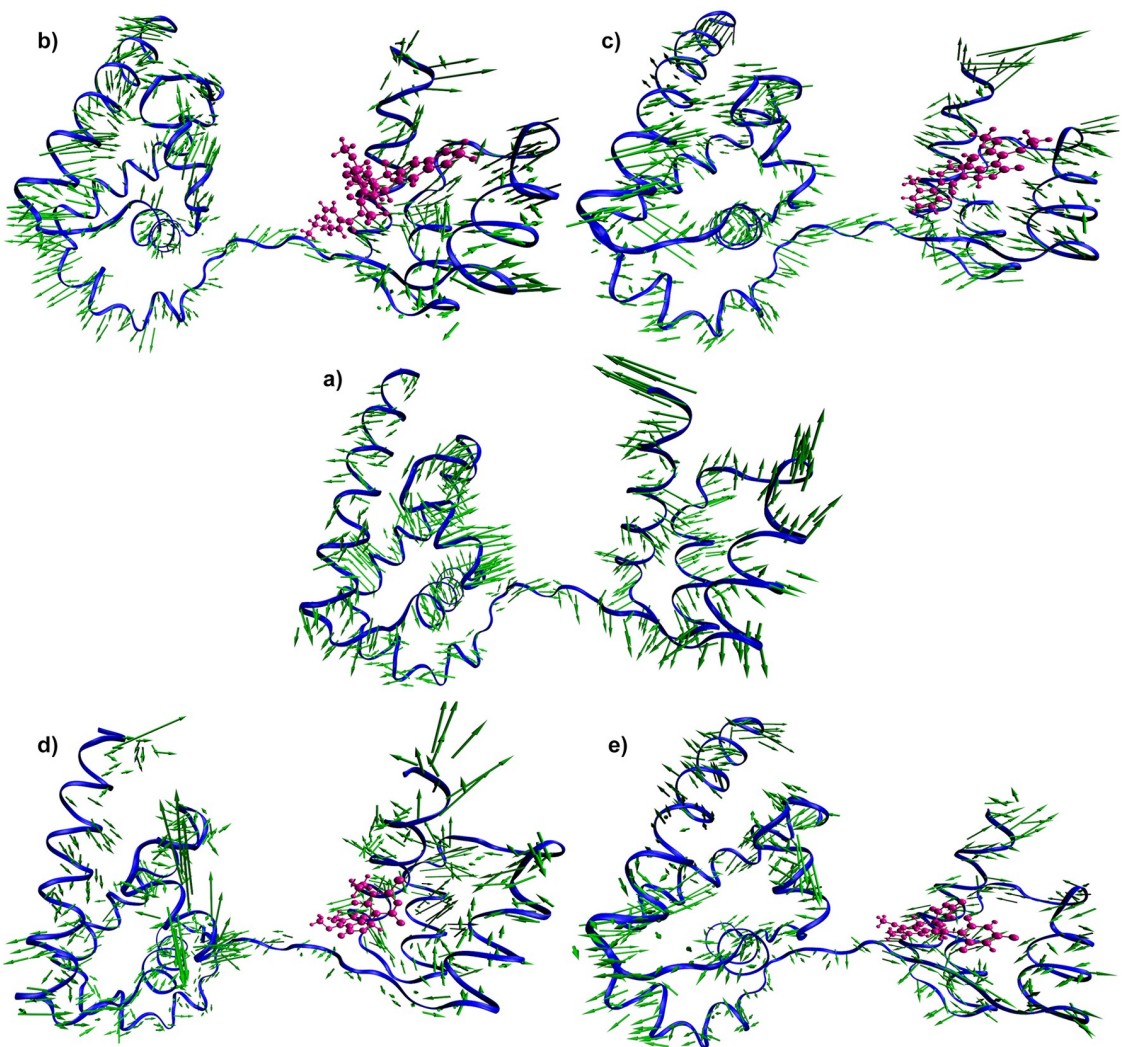

**Fig 11. Mode vector analysis.** Porcupine plots depicting the dominant motions exhibited by the backbone atom of apoprotein (**a**) and the compounds NPB7083 (**b**), ZINC20410920 (**c**), ZINC02145000 (**d**), and ZINC32124203 (**e**) bound proteins. The arrows attached to each backbone atom indicate the direction of movement of protein and the length of each arrow shows the magnitude of the motions.

the simulation. From Fig 11, it was also evidenced that there was a minimal contribution of motions from the $POU_{HD}$ domain after the binding with compounds. More specifically, protein bound to the compounds ZINC32124203 and ZINC02145000 experienced negligible movement in both domains ($POU_S$ and $POU_{HD}$) and also linking region. Particularly, only the C-terminal region and some parts of the POUs loop region showed more flexibility in the ZINC02145000 complex. From this study, we speculated that compound ZINC32124203 and ZINC02145000 may have possessed some communication with the POUs domain rather than only with the $POU_{HD}$ domain.

Structural differences between the five systems were further reviewed where we inspected dynamic cross-correlated motions (DCCM) in the individual systems. The DCCM depicts total correlated motions between the protein residues, where the strongly correlated motions among specific residues are indicated by red color and highly anti-correlated residues are

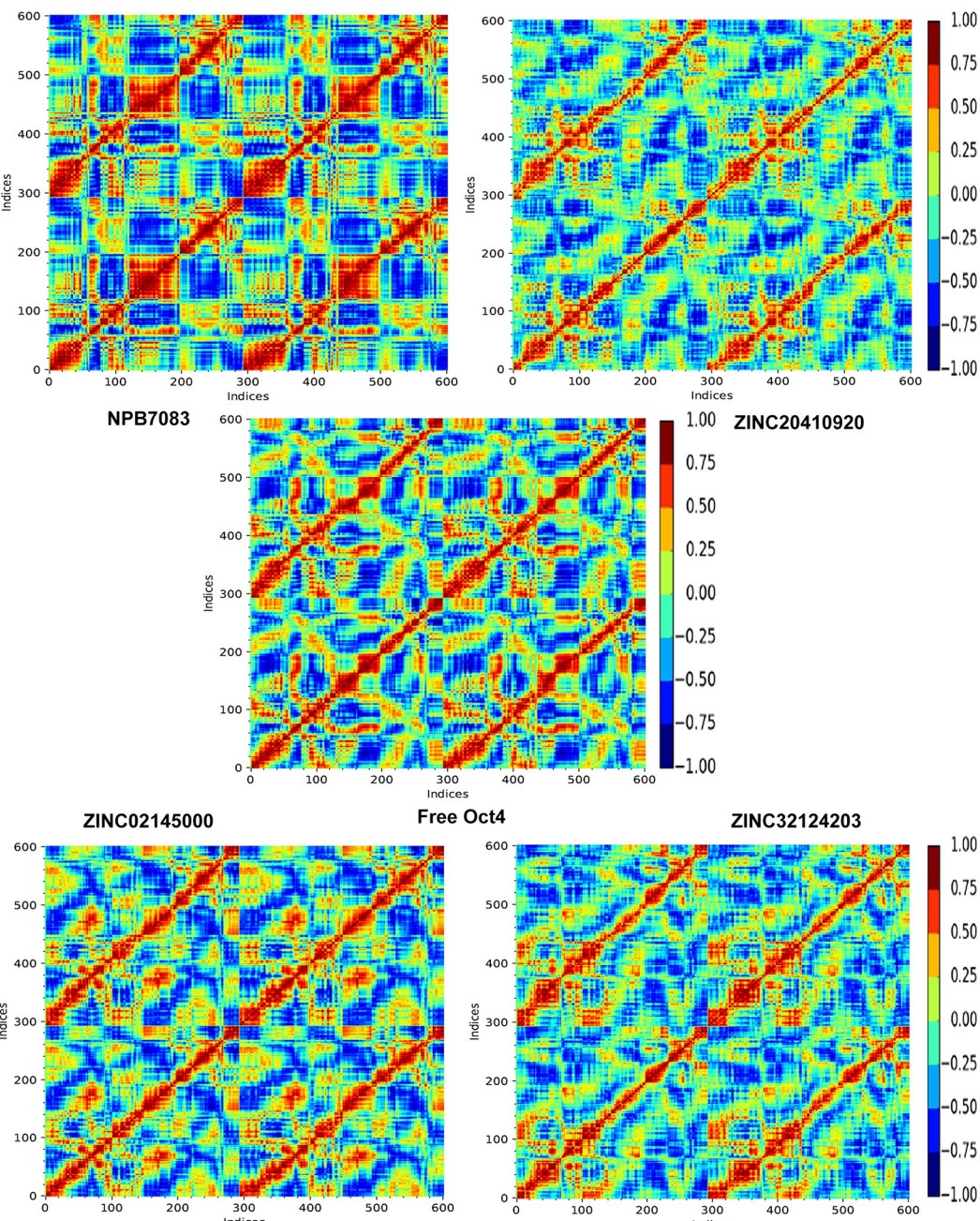

**Fig 12. Dynamic Cross-correlation Matrix (DCCM) of the backbone atoms during100 ns simulations.** The range of the correlated and anti-correlated motions indicated in various colors in the panel where red color denotes positive correlation and blue color specified negative correlation.

specified by blue color (**Fig 12**). As evident in Fig 12, the global dynamics of the apoprotein system were disparities to that of all compound bound complexes, implying that substrate binding imposed some significant secondary structure alterations to the protein which was verified through SSE analysis. However, the ZINC20410920 system showed more no-correlation motions between the amino acids in contrast to other complex systems. Moreover,

NPB7083 exhibited a combination of highly correlated and anti-correlated motions whereas highly correlated motions than anti-correlation motions were observed in ZINC32124203 and ZINC02145000 complexes. Overall, PCA and DCCM analysis explored that the binding of compounds ZINC32124203 and ZINC02145000 decrease the flexibility and increase the correlation motion.

But still, our question remained unanswered why NPB7083 and ZINC20410920 possessed high deviation while ZINC32124203 and ZINC02145000 showed stable conformation. Further molecular dynamics visual inspection unveiled that ZINC32124203 and ZINC02145000 established connections with the POU$_{HD}$ domain as well as POU$_S$ domain while NPB7083 failed to make any communication with the POUs domain and also verified from H-bond analysis. Then we examined all the complexes and found that the compound which formed any interaction with the POUs domain showed stable conformation such as NPB4533, NSC83439, and NSC292567 (high RMSD value but stable conformation). This may be due to the open space provided by simulation while Glide provides a constrained space for docking. From RMSD results, we concluded that a high degree of deviation of the proteins might have come due to considering the two domains of protein along with the linker region for simulation rather than considering only the POU$_{HD}$ domain. From interaction analysis, we found that more interaction with POU$_{HD}$ and POU$_S$ domain more was the stability. From PCA and DCCM analysis, we evidenced that the binding of compounds ZINC32124203 and ZINC02145000 minimized the flexibility and escalated the correlations between the amino acid residues of Oct4. More interestingly, from the current investigation, we anticipated that all compounds are active compounds which can block the activity of the POU$_{HD}$ domain of Oct4 but two compounds (ZINC32124203 and ZINC02145000) having some extraordinary power to block both domain functions.

## Conclusion

Oct4, a master regulator of stem cell maintenance can induce pluripotency in somatic cells which cannot be substituted by any paralogous family member. Alone or in cooperation with Sox2, Nanog and other members, Oct4 activates both protein-coding genes and noncoding RNAs necessary for pluripotency and self-renewal of glial stem cells reprogramming glioblastoma and other cancers. The overexpression of Oct4 contributes to the presence of undifferentiated cells (GSCs) with self-renewal and tumorigenic potential that lead to tumor initiation, invasion, post treatment relapse, and therapeutic resistance. Therefore, we strive to abolish the activity of Oct4 by identifying natural product small molecule inhibitors using computational approaches. Initially, we analyzed the whole sequence to predict the order and disordered regions which demonstrated that Oct4 consists of more disordered regions than the ordered regions and possessing protein-protein interaction regions important for the regulation of GSCs. The order region POU domain containing two subdomains POU$_S$ and POU$_{HD}$ is the most pivotal region to induce pluripotency in somatic cells which bind with their respective half motif DNA independently. Subsequently, we endeavored to examine the protein-protein interaction which depicted that Oct4 mostly interacting with those proteins which are responsible for the maintenance of stem cells, self-renewal, proliferation, pluripotency and lineage differentiation. It is also interacting with some proteins which regulate the expression of VEGF for developing the new blood vessels. Furthermore, to screen small molecule inhibitors targeting the POU$_{HD}$ domain of Oct4 we pre-filtrated three natural product databases (ZincNPD, NCINPD, and NPB) to discriminate the druggable molecules and applied receptor-based virtual screening protocol. Receptor based virtual screening delivered 30 compounds having good docking scores and docking energy. Through visual investigation of interaction,

13 compounds were selected for further analysis. Moreover, the correlation between docking energy and binding free energy provided a significant correlation with 13 ligand efficiency. Furthermore, the conformational stability of the protein after ligand binding was evaluated through molecular dynamics simulation. The simulation results unveiled two compounds ZINC02145000 and ZINC32124203 showing stable backbone and strong interaction with Oct4 even in the presence of linker region and POUs domain and in dynamic condition. More interestingly, ZINC02145000 and ZINC32124203 compounds engaged the POU$_{HD}$ domain as well as POUs domain evidenced from PCA analysis indicating the credible potency against Oct4 function. Lastly, our examination delivered two effective compounds i.e. ZINC02145000 and ZINC32124203 which can impede the Oct4 function and act as anti-GBM or anti-cancer drugs to treat GBM as well as various types of cancer patients with fewer side effects. These new compounds can also show better results in *in vitro* and *in vivo* validation.

## Supporting information

**S1 Fig. Structural comparative analysis between model Oct4 protein (left) and template protein (right) (PDB ID: 3L1P). a)** Secondary structure; **b)** ERRAT analysis; **c)** ProSA knowledge based energy analysis of both model structure and template structure.
(TIF)

**S2 Fig. LigPlot generated 2D diagram of eight screened compounds displaying the hydrogen bond in green color dotted line and hydrophobic bond in red color dotted line.**
(TIF)

**S3 Fig. Conformational stability of selected best four complexes during the 50ns of MD simulation. a)** Represents backbone RMSD of complexes relative to the starting complexes; **b)** Displays RMSF plot of C-alpha fluctuations of selected complexes.
(TIF)

**S4 Fig. Time evaluation of secondary structure element in eight complexes.**
(TIF)

**S5 Fig. Hydrogen bond occupancy by the protein during simulation time period.**
(TIF)

**S1 File. Supporting tables: S1 and S2 Tables.**
(DOCX)

## Acknowledgments

CN and SKS thank Alagappa University for providing the necessary infrastructure.

## Author Contributions

**Conceptualization:** Sanjeev Kumar Singh.

**Data curation:** Chirasmita Nayak.

**Formal analysis:** Chirasmita Nayak.

**Investigation:** Sanjeev Kumar Singh.

**Methodology:** Chirasmita Nayak.

**Project administration:** Sanjeev Kumar Singh.

**Resources:** Sanjeev Kumar Singh.

**Software:** Chirasmita Nayak.

**Supervision:** Sanjeev Kumar Singh.

**Validation:** Chirasmita Nayak.

**Visualization:** Chirasmita Nayak.

**Writing – original draft:** Chirasmita Nayak, Sanjeev Kumar Singh.

**Writing – review & editing:** Chirasmita Nayak, Sanjeev Kumar Singh.

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
