## [Decision Letter · Decision Letter 0]

8 Feb 2021

PONE-D-21-00233

In silico identification of natural product inhibitors against Octamer-binding transcription factor 4 (Oct4) to impede the mechanism of glioma stem cells

PLOS ONE

Dear Dr. Singh,

Thank you for submitting your manuscript to PLOS ONE. After careful consideration, we feel that it has merit but does not fully meet PLOS ONE’s publication criteria as it currently stands. Therefore, we invite you to submit a revised version of the manuscript that addresses the points raised during the review process.

We look forward to receiving your revised manuscript.

Kind regards,

Jie Zheng, Ph.D

Academic Editor

PLOS ONE

"The authors would like to thank the Department of Biotechnology (DBT), Ministry of Science and

Technology [Project Number: BT/PR8138/BID/7/458/2013], Government of India; DST-PURSE 2nd phase

programme order No. SR/PURSE Phase 2/38; FIST (SR/FST/LSI-667/2016); MHRD RUSA 2.0 [F.24-51/2014-

U] for the intramural fund."

Reviewers' comments:

Reviewer's Responses to Questions

**Comments to the Author**

1. Is the manuscript technically sound, and do the data support the conclusions?

Reviewer #1: Partly

2. Has the statistical analysis been performed appropriately and rigorously? 

Reviewer #1: I Don't Know

3. Have the authors made all data underlying the findings in their manuscript fully available?

Reviewer #1: No

4. Is the manuscript presented in an intelligible fashion and written in standard English?

Reviewer #1: Yes

5. Review Comments to the Author

Reviewer #1: This work screened some nature product inhibitors against Oct4 using virtual screening, docking, and MD simulation methods. Two compounds (ZINC02145000 and ZINC32124203) showed a stable and interaction with Oct4, indicating a potential inhibition ability of the two compounds against Oct4. But some issues should be addressed.

(1) Figure 2 is hard to distinguish the interaction network of Oct4, the interaction map combining with a table would be a better to understand the interaction network of Oct4.

(2) There almost no difference between modeled protein and the template protein from Figure 3d. The two figures are exactly same.

(3) What is the accurate simulation time (50 ns in method section, 100 ns in result section)? From Figure 4a, the protein seems to haven’t reach to a equilibrium state, with great structural perturbation even from last 20 ns.

(4) In Figure 4d and Figure 9 for second structure analysis, comparison of the ratio of the secondary structures between initial simulation time and equilibrium state is better to understand the structure transition of the protein.

(5) In Figure 5, a close-up representation of specific binding residues of the Oct4 would be better to address the binding site.

6. PLOS authors have the option to publish the peer review history of their article (what does this mean?). If published, this will include your full peer review and any attached files.

Reviewer #1: No

---

## [Author Response · Author response to Decision Letter 0]

7 Jul 2021

Response to Decision Letter and Reviewer Comments

We would like to thank the editors and reviewers for their constructive suggestions and comments to improve the quality and shape of our manuscript entitled “In silico identification of natural product inhibitors against Octamer-binding transcription factor 4 (Oct4) to impede the mechanism of glioma stem cells”. We believe our refined manuscript is improved based on the reviewer suggestions. With this, we have addressed the each of their concerns as outlined below. 

Note:

The reviewer’s comments are in black and italics.

Author Responses are in red and bold text.

Changes in the manuscript were highlighted within the manuscript (Ms) in red colour.

Response to Reviewer #1: 

Comment 1:

 Figure 2 is hard to distinguish the interaction network of Oct4; the interaction map combining with a table would be a better to understand the interaction network of Oct4.

We would like to thank the Reviewer for careful reading of this Ms and for the comment. As per the reviewer suggestion, the tabulated representation of interaction network of Oct4 has been added in the manuscript and highlighted in red color (Table 1).

Comment 2:

There almost no difference between modeled protein and the template protein from Figure 3d. The two figures are exactly same.

We agree with the reviewer that our modeled protein structure is almost similar with template structure. There were only some differences in loop regions so they both superimposed perfectly with RMSD of 0.102Å which was indicating that the generated model is more reliable and accurate to perform further analysis. 

Comment 3:

What is the accurate simulation time (50 ns in method section, 100 ns in result section)? From Figure 4a, the protein seems to have not reached to an equilibrium state, with great structural perturbation even from last 20 ns.

We would like to thank reviewer for careful reading of this Ms. To address your query, we have performed 100ns molecular dynamics for apo protein while for complexes 50ns.

We agree with reviewer. Since after 50ns, apo protein has not shown any stability during simulation then we have extended to 100ns time period. However after 100ns also the apo protein was exhibited high deviation and instability throughout the simulation then we try to find out the actual reason for the same through literature survey. We found an article publish in PLoS One (ref. No. 86) which has been discussed in the MS where they have reported that oct4 was highly unstable after insertion of only 3 base pairs in oct4 at oct4/sox2-DNA binding site which roughly packed protein arrangement and weaken the interactions between the Oct4 and Sox2 proteins. From this we have hypothesized that maybe oct4 will show stable conformation after binding with small molecule inhibitors. The hypothesis was confirmed by performing molecular dynamic simulation of complexes which disclosed that after binding with our screened compounds oct4 was stable even within 35ns. Hence we have not extended our dynamics to 100ns for complex proteins.

Comment 4:

In Figure 4d and Figure 9 for second structure analysis, comparison of the ratio of the secondary structures between initial simulation time and equilibrium state is better to understand the structure transition of the protein.

As suggested by the reviewer, we have added one table stating the ratio of SSE of initial structure (apo protein) and compound binding proteins which has been highlighted in red color (Table 5). 

Comment 5:

In Figure 5, a close-up representation of specific binding residues of the Oct4 would be better to address the binding site.

As suggested by the reviewer, a closer representation of binding site with its amino acids has been induced in the figure 5 and incorporated in the manuscript.

---

## [Editor Report · Decision Letter 1]

26 Jul 2021

In silico identification of natural product inhibitors against Octamer-binding transcription factor 4 (Oct4) to impede the mechanism of glioma stem cells

PONE-D-21-00233R1

Dear Dr. Singh,

We’re pleased to inform you that your manuscript has been judged scientifically suitable for publication and will be formally accepted for publication once it meets all outstanding technical requirements.

Kind regards,

Jie Zheng, Ph.D

Academic Editor

PLOS ONE
---

## [Editor Report · Acceptance letter]

3 Aug 2021

PONE-D-21-00233R1 

*In silico* identification of natural product inhibitors against Octamer-binding transcription factor 4 (Oct4) to impede the mechanism of glioma stem cells 

Dear Dr. Singh:

I'm pleased to inform you that your manuscript has been deemed suitable for publication in PLOS ONE. Congratulations! Your manuscript is now with our production department. 

Kind regards, 

on behalf of

Dr. Jie Zheng 

Academic Editor

PLOS ONE